# A novel bivalent interaction mode underlies a non-catalytic mechanism for Pin1-mediated protein kinase C regulation

Xiao-Ru Chen[1†], Karuna Dixit[1†], Yuan Yang[1], Mark I McDermott[2], Hasan Tanvir Imam[1], Vytas A Bankaitis[2], Tatyana I Igumenova[1,2]*

[1]Department of Biochemistry & Biophysics, Texas A&M University, College Station, United States; [2]Department of Cell Biology & Genetics, Texas A&M University, College Station, United States

*For correspondence:
tigumenova@tamu.edu

[†]These authors contributed equally to this work

Competing interest: The authors declare that no competing interests exist.

**Abstract** Regulated hydrolysis of the phosphoinositide phosphatidylinositol(4,5)-bis-phosphate to diacylglycerol and inositol-1,4,5-$P_3$ defines a major eukaryotic pathway for translation of extracellular cues to intracellular signaling circuits. Members of the lipid-activated protein kinase C isoenzyme family (PKCs) play central roles in this signaling circuit. One of the regulatory mechanisms employed to downregulate stimulated PKC activity is via a proteasome-dependent degradation pathway that is potentiated by peptidyl-prolyl isomerase Pin1. Here, we show that contrary to prevailing models, Pin1 does not regulate conventional PKC isoforms α and βII via a canonical *cis-trans* isomerization of the peptidyl-prolyl bond. Rather, Pin1 acts as a PKC binding partner that controls PKC activity via sequestration of the C-terminal tail of the kinase. The high-resolution structure of full-length Pin1 complexed to the C-terminal tail of PKCβII reveals that a novel bivalent interaction mode underlies the non-catalytic mode of Pin1 action. Specifically, Pin1 adopts a conformation in which it uses the WW and PPIase domains to engage two conserved phosphorylated PKC motifs, the turn motif and hydrophobic motif, respectively. Hydrophobic motif is a non-canonical Pin1-interacting element. The structural information combined with the results of extensive binding studies and experiments in cultured cells suggest that non-catalytic mechanisms represent unappreciated modes of Pin1-mediated regulation of AGC kinases and other key enzymes/substrates.

## Editor's evaluation

Pin1 is an essential prolyl cis/trans isomerase which has attracted considerable attention as this enzyme family is implicated in cancer and neurodegenerative diseases. Here, authors provide compelling evidence that Pin1 modulates the activity of an important cell signaling kinase, Protein Kinase C by a non-catalytic mechanism. This result suggests a new mode of Pin1 action, regulating the cellular stability of the kinase via a chaperone-like activity.

## Introduction

Protein kinase C isoenzymes (PKCs) define a family of multi-modular Ser/Thr kinases that regulate cell growth, differentiation, apoptosis, and motility (*Clemens et al., 1992*; *Rosse et al., 2010*). PKCs occupy a key node of the phosphoinositide signaling pathway whose intracellular arms are mediated by diacylglycerol (DAG) and IP$_3$-dependent Ca$^{2+}$ signaling (*Newton, 2010*). Novel and conventional PKCs are allosterically activated upon translocation to membranes. This membrane recruitment

process involves PKC interactions with DAG and, in the case of $Ca^{2+}$-dependent isoforms, the anionic phospholipids phosphatidylserine and PtdIns(4,5)$P_2$ (*Newton and Protein kinase, 1995*; *Johnson et al., 2000*; *Evans et al., 2006*). Dysregulated PKC activity is implicated in cancer progression (*Isakov, 2018*; *Cooke et al., 2017*), cardiac disease (*Singh et al., 2017*), diabetes (*Rask-Madsen and King, 2005*), and neurodegenerative disorders (*Lordén and Newton, 2021*). Both gain- and loss-of function PKC mutations are associated with diseased states, as reported recently for certain cancers (*Antal et al., 2015b*; *Parker et al., 2021*) and neurodegenerative disorders (*Alfonso et al., 2016*; *Callender et al., 2018*).

Central to the question of PKC regulation is its phosphorylation state as it determines both cellular steady-state levels of the enzyme and its enzymatic activity. During the maturation process, a sequence of four ordered phosphorylation events (*Baffi et al., 2021*) enable PKC to adopt a stable autoinhibited conformation in the cytosol. The autoinhibitory interactions between the N-terminal pseudo-substrate region and the C-terminal kinase domains are allosterically released by the interactions of second messengers with the PKC regulatory domain (*Figure 1A*). The activated kinase can be additionally stabilized through the interactions with RACK (receptors for activated C-kinase) adaptor proteins (*Mochly-Rosen et al., 1991*; *Stebbins and Mochly-Rosen, 2001*) that play an important role in the subcellular localization of PKCs (*Ron et al., 1999*). The open conformation that PKC assumes upon activation makes it susceptible to dephosphorylation by the protein phosphatase 2A (PP2A) (*Srivastava et al., 2002*) and pleckstrin homology domain and leucine rich repeat protein phosphatase (PHLPP) phosphatases (*Gao et al., 2008*; *Baffi et al., 2019*) with the result that the enzyme is rapidly degraded in the cell. This 'activation-induced' downregulation is one of the major mechanisms for terminating the PKC-mediated signaling response. Pin1, a peptidyl-prolyl isomerase, plays an important role in that process. It is for this reason that Pin1 was coined as the 'molecular timer' for PKC lifetime (*Abrahamsen et al., 2012*).

Pin1 is a peptidyl-prolyl isomerase of the parvuline family and is unique in its specificity toward pSer/pThr-Pro protein motifs (*Lu et al., 1996*). These Ser/Thr-Pro motifs constitute ~30% of all phosphorylation sites in the proteome and their phosphorylation is catalyzed by proline-directed kinases (*Ubersax and Ferrell, 2007*). It is through isomerization of the *cis-trans* pSer/pThr-Pro bond that Pin1 brings the associated conformational changes of its substrates into biologically relevant timescales. In turn, conformation-specific activities of Pin1 client proteins are essential for the regulation of many cellular processes including metabolism, cell cycle progression, apoptosis, cell motility, cell proliferation, and cell survival (*Liou et al., 2011*). Pin1 is overexpressed/overactivated in cancers with the result that numerous oncoproteins are activated and tumor suppressor functions are deactivated (*Lu and Hunter, 2014*; *Zhou and Lu, 2016*; *Zannini et al., 2019*; *Chen and Igumenova, 2023*). It is because Pin1 stimulates oncogenic pathways that Pin1 inhibitors show great promise in the development of new cancer therapies (*Moore and Potter, 2013*; *Pinch et al., 2020*; *Wei et al., 2015*; *Campaner et al., 2017*; *Dubiella et al., 2021*; *Koikawa et al., 2021*; *Kozono et al., 2018*).

Pin1 consists of two domains – the WW and PPIase modules (*Ranganathan et al., 1997*). The interplay between these domains is relevant to Pin1 catalytic function as it ensures the adaptability of Pin1 to a variety of phosphorylated substrates. Both Pin1 domains possess structural elements capable of interacting with pSer/pThr-Pro motifs, but only the isomerase domain has a catalytic role in the peptidyl-prolyl bond isomerization. The WW domain exhibits an affinity for Pin1 substrates that is ~10-fold greater than that of the PPIase domain (*Lee and Liou, 2018*), and the WW domain is thought to either facilitate substrate recruitment to the PPIase active site and/or preferentially stabilize binding of a particular substrate isomer (*Lu et al., 2002*; *Zhou et al., 1999*; *Verdecia et al., 2000*). The linker connecting the two domains confers significant flexibility to the Pin1 structure, and the conformational ensemble of Pin1 in solution is comprised of an ~70:30 ratio of 'compact' to 'extended' conformers (*Born et al., 2021*). Thus, Pin1 dynamics at both inter- and intra-domain levels play essential roles in the allosteric behavior of the enzyme (*Peng, 2015*; *Guo et al., 2015*).

The current model of PKC regulation by Pin1 is based on extensive cell biological evidence. It posits that Pin1 catalyzes a *cis-trans* isomerization of the C-terminus of the conventional (or $Ca^{2+}$-dependent) PKC isoforms α and βII (*Abrahamsen et al., 2012*). Herein, we report that, contrary to the prevailing model of Pin1 action, Pin1's role in PKC α and βII regulation is a non-catalytic one. Instead, Pin1 acts as a PKC binding partner that sequesters two conserved phosphorylated PKC motifs in the disordered C-terminal tail (C-term) of the kinase. Our structure of the Pin1-C-term PKCβII complex

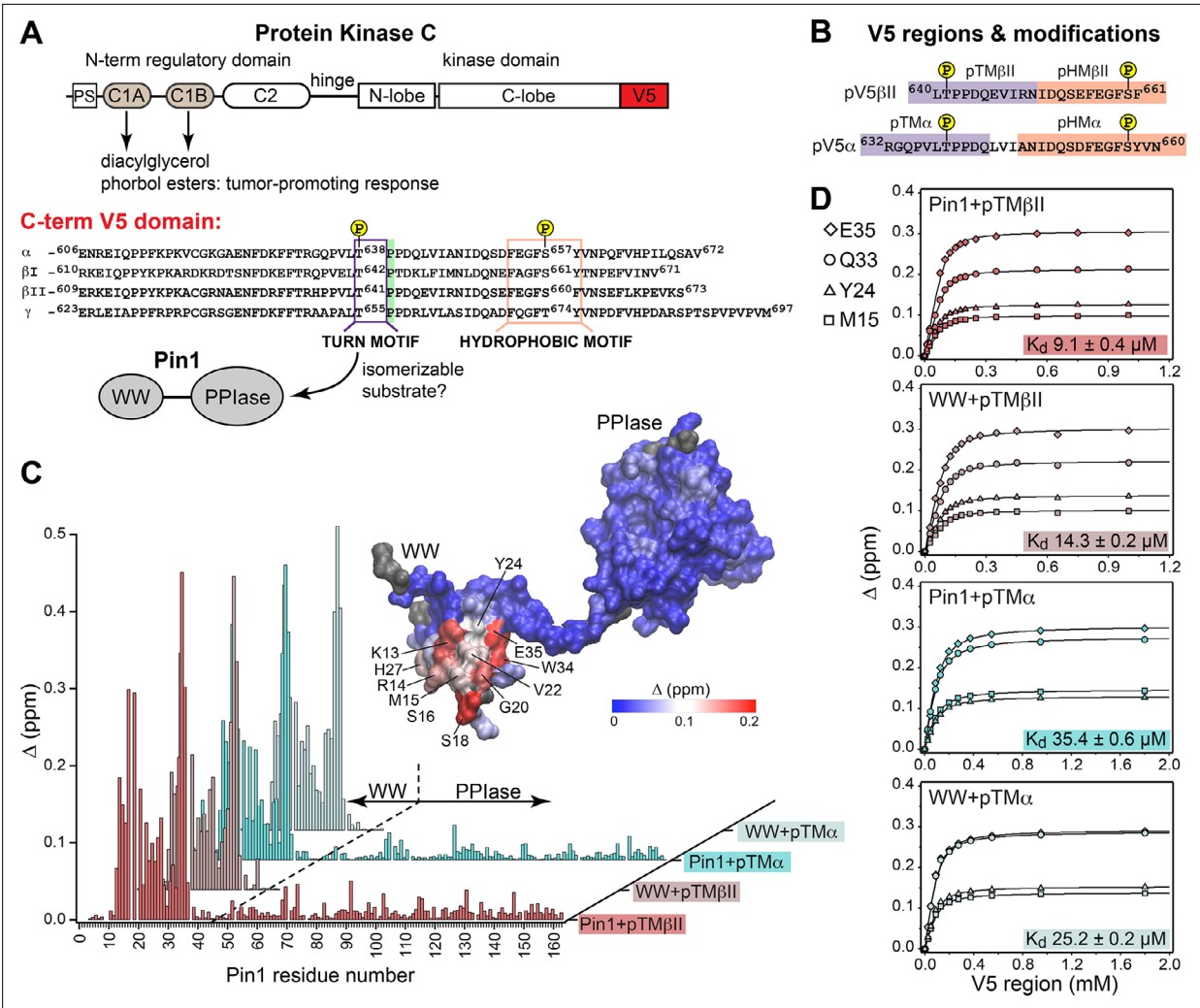

**Figure 1.** Pin1 binds the turn motifs of α and βII PKC isoenzymes through its WW domain. (**A**) Multi-modular architecture of conventional PKC isoenzymes, shown along with the amino acid sequence alignment of the C-terminal V5 domains. Turn and hydrophobic motifs are highlighted in purple and orange, respectively. (**B**) Notations for the V5 peptides that were selected for the NMR-detected binding experiments conducted in this study. (**C**) Residue-specific chemical shift perturbation (CSP) plots of Pin1 and its isolated WW domain obtained upon binding of pTMβII and pTMα. The turn motifs (TMs) interact exclusively with the WW domain. The CSP values Δ for the Pin1::pTMβII complex are color-coded and mapped on to lowest-energy solution NMR structure of apo Pin1 (PDB: 1NMV). (**D**) Representative binding isotherms and dissociation constants for the formation of the complexes between TMs and Pin1 and its WW domain. The source 2D $^{15}$N-$^{1}$H heteronuclear single-quantum coherence (HSQC) NMR spectra are given in *Supplementary files 1–4*.

The online version of this article includes the following figure supplement(s) for figure 1:

**Figure supplement 1.** NMR-detected binding of the PKCβII turn motif (pTMβII) to full-length Pin1.

**Figure supplement 2.** NMR-detected binding of the PKCβII turn motif (pTMβII) to the isolated WW domain.

**Figure supplement 3.** NMR-detected binding of the PKCα turn motif (pTMα) to full-length Pin1.

**Figure supplement 4.** NMR-detected binding of the PKCα turn motif (pTMα) to the isolated WW domain.

**Figure supplement 5.** Mass spectrometry data for the synthesized peptides derived from the C-terminal V5 regions of PKC isozymes.

reveals a novel bivalent interaction mode that has not been previously observed for Pin1, and is the first structure of the full-length Pin1 complexed to a ligand that engages both domains. Our structural and biophysical data provide the molecular basis of non-catalytic Pin1 action that is also supported by the results of activity assays and experiments in the cellular context.

## Results

### The turn motif of the α and βII PKC isoenzymes preferentially binds to the WW domain of Pin1

The key feature of the current model for Pin1-mediated PKC regulation is the Pin1-catalyzed *cis-trans* isomerization of the turn motif (TM) (*Figure 1A*). TM is a conserved feature among many AGC kinases that presents a phosphorylatable Ser or Thr residue in the C-terminal domains of these enzymes (*Pearce et al., 2010*). Phosphorylation of the TM is part of the PKC maturation process, and it is essential for enzyme stability (*Newton, 2010*; *Gao and Newton, 2002*). In conventional PKC isoforms (α, βI/II, γ), the TM is followed by a proline residue and this pThr-Pro motif is proposed to be the isomerizable Pin1 target (*Zhou et al., 1999*; *Figure 1A*). Therefore, the first step toward understanding how Pin1 regulates PKC was to determine the interaction mode between Pin1 and TM. To that end, NMR-detected binding experiments were conducted using uniformly $^{15}$N-enriched ([U-$^{15}$N]) WW domain and full-length Pin1, and two synthetic peptides (pTMα and pTMβII) that correspond to the phosphorylated TMs of the α and βII PKC isoforms, respectively (*Figure 1B*). Addition of increasing concentrations of pTMα and pTMβII resulted in drastic shifts in the 2D $^{15}$N-$^{1}$H heteronuclear single-quantum coherence (HSQC) spectra of [U-$^{15}$N] Pin1 and WW (*Figure 1—figure supplements 1–4*). The residue-specific chemical shift perturbation (CSP) values were calculated for pairs of spectra corresponding to the maximum concentrations of pTM in each titration series *vs.* apo Pin1. The four CSP plots of *Figure 1C* clearly demonstrate that the pTM binding site resides on the WW domain, and that the interaction mode is not significantly influenced by the presence of the PPIase domain in full-length Pin1. The fast exchange regime of the pTM-protein interactions enabled us to construct binding curves for all resolved residues with CSPs larger than the mean (see "Construction and analysis of the chemical shift-based binding curves" in the Methods section for details on the selection criteria and fitting procedures). The curves fit well globally with a single-site binding model (*Figure 1D*, *Supplementary file 2*). The reported global $K_d$ values fall into the micromolar range typical for WW-substrate affinities (*Verdecia et al., 2000*; *Born et al., 2019*; *Wilson et al., 2013*). The similarities of the global $K_d$ values between WW and full-length Pin1 (*Figure 1D*) further support the conclusion that the PPIase domain does not significantly affect the pTM-WW interactions, and that the TMs of the α and βII PKC isoforms bind preferentially to the Pin1 WW domain.

### Pin1 does not catalyze the *cis-trans* isomerization of TM in α and βII PKC isoenzymes

The lack of high-affinity interactions of the PPIase domain with substrate does not preclude a catalytic mode of action for Pin1 as demonstrated previously for several Pin1 substrates (*Wang et al., 2015a*). We therefore tested the ability of Pin1 to isomerize the TM. These experiments monitored activity against the isolated TM and in the context of a larger C-terminal region that harbors a second phosphorylated motif found in most AGC kinases – the hydrophobic motif (HM). HMs in PKC isoforms are described by the sequence FXXF(S/T)(F/Y) where the underlined Ser/Thr is constitutively phosphorylated as part of the kinase maturation process (*Newton, 2010*; *Edwards and Newton, 1997*). HM is located downstream of the TM with 14 amino acid residues separating the two motifs. Cell biological studies suggest that the HM is involved in PKC interactions with Pin1 (*Abrahamsen et al., 2012*). However, the functional role of the HM is unclear as it does not fit the definition of a canonical Pin1 substrate due to the absence of a Pro residue that follows pSer/Thr. In these and all subsequent experiments, we used two synthetic double-phosphorylated peptides (pV5α and pV5βII; *Figure 1B*) to model the C-term PKC regions that contain both phosphorylated HM and TM.

Pin1 catalytic activity is readily measured using the $^{1}$H-$^{1}$H exchange spectroscopy (EXSY) of substrates in the presence of catalytic amounts of the enzyme (*Smet et al., 2005*; *Jinasena et al., 2019*; *Mercedes-Camacho et al., 2013*). EXSY experiments are applicable to characterization of chemical exchange processes with the exchange rates between 0.1 s$^{-1}$ and 100 s$^{-1}$ (*Palmer et al., 2001*; *Kawale and Burmann, 2023*). Pin1 brings the slow *cis-trans* isomerization process of pSer/Thr-Pro bonds into biologically relevant timescales by enhancing the isomerization rate ~10$^{3}$- to 10$^{4}$-fold. This rate enhancement gives rise to characteristic cross-peaks in the 2D $^{1}$H-$^{1}$H EXSY NMR spectra. To our surprise, addition of catalytic amounts of Pin1 to either isolated pTMβII or to the entire phosphorylated pV5βII region failed to generate cross-peaks between the amide $^{1}$H resonances of

pThr641 in the *trans*(t) and *cis*(c) conformations of the pThr641(–1)-Pro642(0) bond in the pThr641(–1)-Pro642(0)-Pro643(+1) segment (*Figure 2A*). We intentionally used a long mixing time (0.5 s) to enable the detection of slow processes. Yet, no evidence of significant rate enhancement was observed, suggesting that even upon addition of Pin1 the exchange rate between the isomers remains slower than 0.1 s$^{-1}$.

The lack of appreciable enhancement of the isomerization rate was similarly obtained for the pThr638(–1)-Pro639(0)-Pro640(+1) segment of the pTMα and pV5α regions (*Figure 2B*). Of note, we were able to detect Pin1-catalyzed *cis-trans* isomerization of the peptidyl-prolyl bond between Gln634 and Pro635 in pTMα. This reaction was manifested by the appearance of cross-peaks between the amide $^1H_N$ of Gln634 in the *cis* and *trans* conformations (*Figure 2B*). Although Gln634-Pro635 is not a canonical Pin1 substrate, the catalytic action of Pin1 brings the reaction rate into a detectable range with $k_{EX}$ of ~1 s$^{-1}$, where $k_{EX}$ is the sum of the forward and reverse kinetic rate constants of the isomerization reaction, $k_{tc}$ and $k_{ct}$. The Gln634-Pro635 data serve as direct evidence that weak interactions of the TM region with the PPIase domain are sufficient for Pin1 to appreciably catalyze the *cis-trans* isomerization of the non-canonical Gln-Pro substrate but not the pThr-Pro segment of the TM. For both PKC α and βII isoforms, the HM has no detectable effect on Pin1 catalytic activity against these substrates.

## The proline residue at the +1 position prohibits Pin1-mediated *cis-trans* isomerization of TM in α and βII PKC isoenzymes

We hypothesized that the proline at position +1 rendered the pTMα and pTMβII motifs poor substrates for PPIase-catalyzed isomerization. Thus, we generated the P640A variant of pTMα by replacing Pro at +1 position with Ala. Pin1-catalyzed isomerization of the pThr638-Pro639 bond in the P640A variant is readily detectable as reported by the appearance of distinct cross-peaks between the *cis* and *trans* conformers (*Figure 2C*). The $k_{EX}$ value of 29.6 s$^{-1}$ obtained from the time dependence of the cross-peak intensities (see "$^1$H-$^1$H exchange spectroscopy" in the Methods section) is typical for the Pin1 substrates. Similarly, replacement of Pro at the (+1) position with Thr in the short LpTPPD peptide common to the TM regions of α, βII, and γ PKC isoforms also resulted in significant enhancement of the *cis-trans* isomerization rate ($k_{EX}$ value of 46.2 s$^{-1}$; *Figure 2D*). We conclude that Pro at the (+1) position of TMs is incompatible with these motifs serving as efficient substrates for isomerization by Pin1.

Our collective data further suggest that, contrary to prevailing models, Pin1 does not control downregulation of PKCα and βII isoforms by catalyzing isomerization of the PKC tail. Rather, Pin1 does so via a non-catalytic mechanism. The novelty of our findings prompted us to more thoroughly investigate the biophysical and structural basis of Pin1 interactions with the C-terminal domains of α and βII PKC isoforms. We focused on three aspects: (i) identifying the role of the HM, (ii) defining the binding mode, and (iii) establishing the effect of the C-terminal domain phosphorylation state on interactions with Pin1. To those ends, we conducted a total of 19 NMR-detected binding experiments between Pin1 and the relevant regions of the C-term domains of PKC α and βII isoforms with different phosphorylation states (*Supplementary file 1*). All information regarding the notations, sequences, and affinities is given in *Supplementary file 2*. The Pin1 domains are identified with the subscript 'iso' and 'Pin1' in their isolated forms and full-length Pin1 contexts, respectively.

## HM of PKC binds to Pin1 via two independent sites

The HM (PKC α/βII) sequence FEGFpSF/Y does not fit the canonical definition of a Pin1 substrate, and its role in the PKC-Pin1 interactions is unknown. Addition of the phosphorylated HM regions, pHMα and pHMβII, to [U-$^{15}$N] full-length Pin1 resulted in large CSPs of the amide N-H$_N$ cross-peaks for many Pin1 residues, thereby providing direct and site-specific evidence for the Pin1-HM interactions (*Figure 3A* and *Figure 3—figure supplement 1A and B*). We were surprised to find that residues in both Pin1 domains were significantly affected, suggesting the existence of more than one pHM binding site in Pin1 (*Figure 3A* and *Figure 3—figure supplement 2A*). To identify which Pin1 domain harbors the HM binding site(s), we titrated isolated [U-$^{15}$N] WW and PPIase domains with pHMβII and α (*Figure 3B* and *Figure 3—figure supplement 2B*). The CSP patterns of full-length Pin1 and the sum of isolated domains are essentially identical. These data indicate that Pin1 contains two sites that bind the phosphorylated HM: one residing in the WW domain and the other in the PPIase domain (*Figure 3A and B* and *Figure 3—figure supplement 2*).

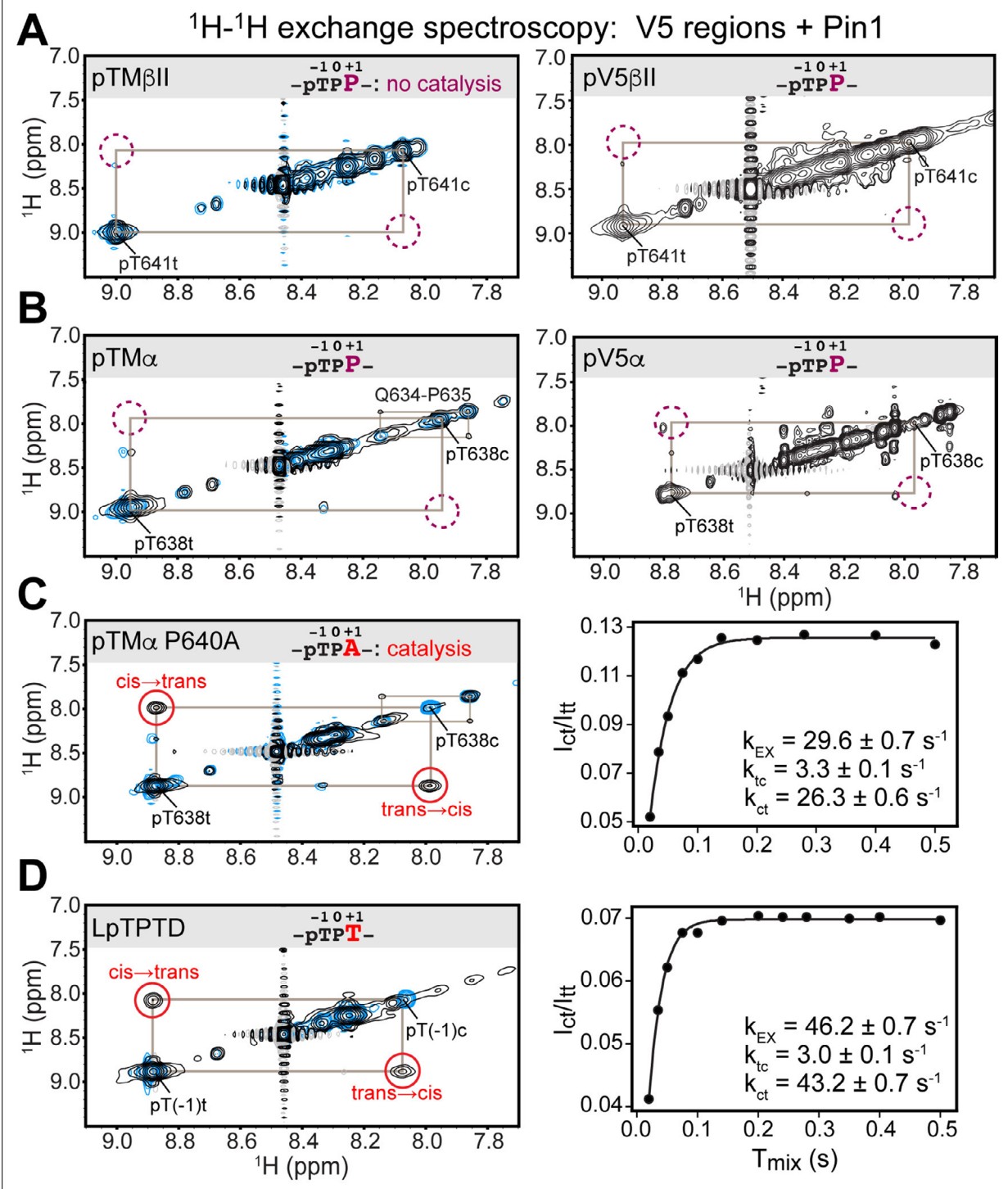

**Figure 2.** Pin1 does not appreciably catalyze isomerization of the turn motif in α and βII PKC isoenzymes due to presence of proline at the +1 position. No exchange cross-peaks characteristic of Pin1-catalyzed pThr-Pro *cis-trans* isomerization are present in the spectra of turn motif (TM) regions from the PKC βII (**A**) and α (**B**) isoforms. This is demonstrated for both, isolated TM and the pV5 regions that harbor both HM and TM. Non-specific catalysis by Pin1 is evident in the appearance of exchange cross-peaks for the $^1H_N$ of Gln634 in *cis* and *trans* conformations of the Gln634-Pro635 segment (**B**). Replacement of Pro640 with Ala at the (+1) position of pTMα results in significant rate enhancement with the $k_{EX}$ value of 29.6 s$^{-1}$ (**C**). Thr at the (+1) position similarly enhances the rate of isomerization, as demonstrated for a short five-residue LpTPPD peptide common to the TM regions of α, βII, and γ PKCs (**D**). The reference spectra collected with same parameters in the absence of Pin1 are shown in blue. The concentration of the V5 region peptides was 1–2 mM, with Pin1 added at catalytic amounts of 50 μM. The mixing times for all spectra are 0.5 s. The NMR spectra show the expansion of the $^1H$-$^1H$ amide region.

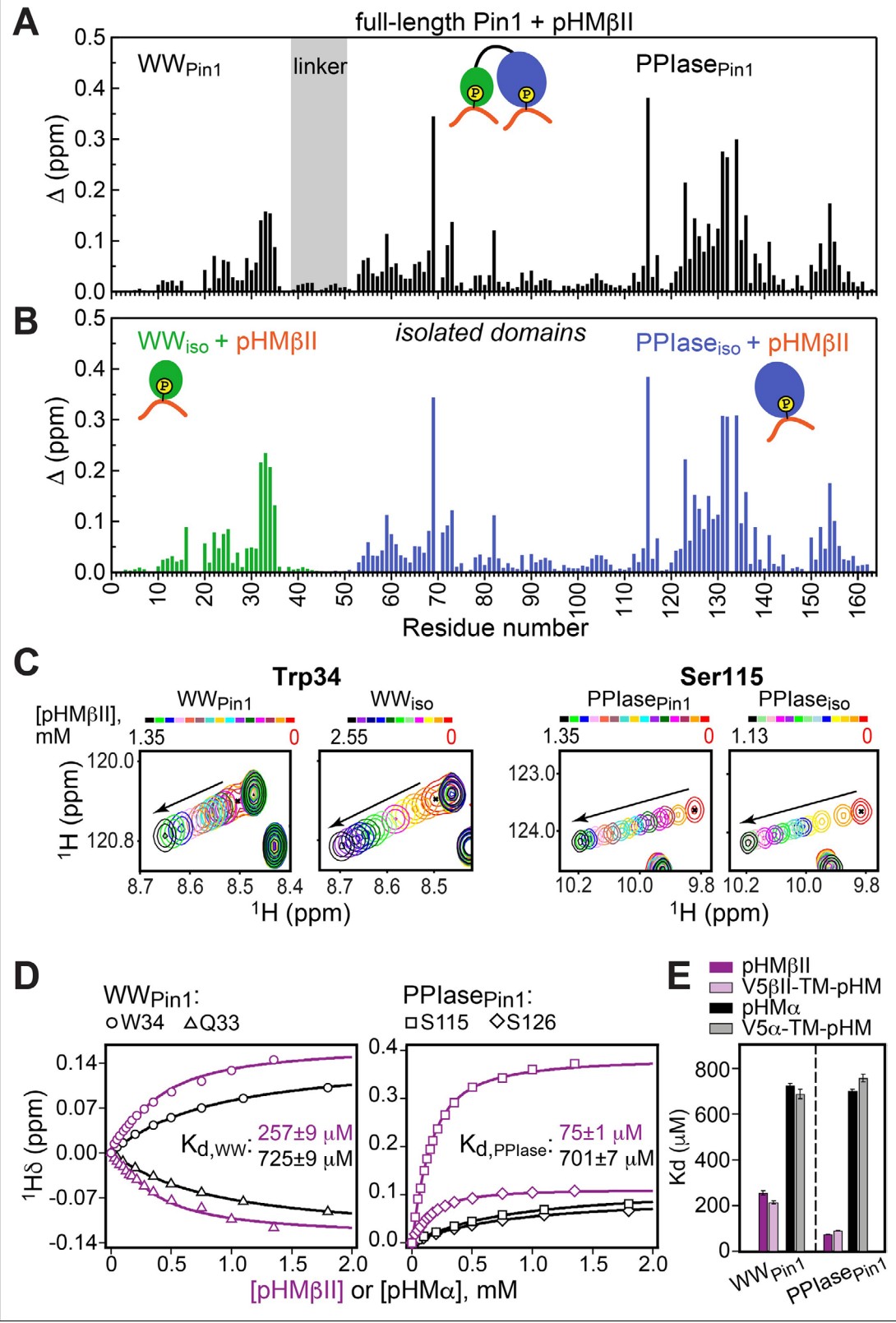

**Figure 3.** Hydrophobic motif interacts with Pin1 via two independent sites. Comparison of the chemical shift perturbation (CSP) plots obtained at maximum concentrations of pHMβII versus ligand-free proteins for (**A**) full-length Pin1 and (**B**) isolated WW and PPIase domains. The similarity of the CSP patterns in (**A**) and (**B**) indicates that pHM has two independent binding sites in Pin1, one per domain. (**C**) Expansion of the $^{15}$N-$^{1}$H chemical shift correlation spectra of Trp34 in the WW domain and Ser115 in the PPIase domain showing the fast-exchange regime of pHM binding. Full spectra are

*Figure 3 continued on next page*

*Figure 3 continued*

given in *Figure 3—figure supplement 1*. (**D**) Representative pHMβII (purple) and pHMα (black) binding curves for the Pin1 residues that belong to the WW and PPIase domains. Solid lines are the global fits to a model with two independent binding sites. (**E**) The unphosphorylated turn motif (TM) region upstream of the hydrophobic motif (HM) has no influence on the HM interaction mode with Pin1. Experimental conditions for all α and βII experiments are given in *Supplementary file 2*, experimental IDs # 5–10, 12, and 13.

The online version of this article includes the following figure supplement(s) for figure 3:

**Figure supplement 1.** NMR-detected binding of hydrophobic motifs from PKCβII (A, pHMβII) and α (B, pHMα) to full-length Pin1.

**Figure supplement 2.** Pin1 binds hydrophobic motif of PKCα (pHMα) via two independent sites.

The fast-exchange regime of pHM binding to Pin1 (*Figure 3C*) enabled us to construct the binding curves for all responsive residues and fit them globally using the two-site binding model ("Construction and analysis of the chemical shift-based binding curves" in the Methods section). The fitting produced domain-specific $K_d$ values within full-length Pin1 (*Figure 3D*). The pHMα motif has comparable affinities to both domains (725 µM to $WW_{Pin1}$ and 701 µM to $PPIase_{Pin1}$, *Supplementary file 2*), whereas the pHMβII motif has a 3.5-fold greater affinity for $PPIase_{Pin1}$ than it does for $WW_{Pin1}$ ($K_d$ values of 75 µM and 257 µM, respectively; *Supplementary file 2*). To determine if the presence of the upstream unphosphorylated TM influences pHM-Pin1 interactions, we tested the binding of the V5α-TM-pHM and V5βII-TM-pHM regions to full-length Pin1. The NMR spectra and the measured $K_d$ values are identical to those for the pHM regions only (*Figure 3E*). Thus, unphosphorylated TM has little effect on the HM interactions with Pin1. The collective data demonstrate that, despite not being a canonical substrate, the phosphorylated HM of PKC isoenzymes can interact with Pin1 via two independent binding sites that reside on the WW and PPIase domains, respectively.

## Pin1 engages in unidirectional high-affinity bivalent interactions with the PKC C-terminus

Both HM and TM are phosphorylated in mature PKC. Addition of the pV5βII and pV5α regions that harbor both phosphorylated motifs to full-length Pin1 produced large CSP values in both Pin1 domains (*Figure 4A*, *Figure 4—figure supplements 1 and 2A*). The CSP plot of full-length Pin1 complexed to pV5βII(α) matches the sum of the CSP plots obtained for the isolated WW domain complexed to pTMβII(α), and the isolated PPIase domain complexed to pHMβII(α) (*Figure 4A and B*; *Figure 4—figure supplements 1 and 2*). These data provide unambiguous evidence that full-length Pin1 engages in bivalent interactions with the PKC C-terminal tail where WW binds the TM and PPIase binds the HM. The directionality of these interactions is imposed by the binding preferences of the TM. As the TM primarily interacts with the WW domain (*Figure 1C*), the only site available for HM binding resides on the PPIase domain. Neither motif is an isomerizable Pin1 substrate.

Since the Pin1-pV5βII interactions fall into the intermediate-to-fast binding regime, NMR lineshape analysis was applied to obtain the $K_d$ value of 1.5 µM ("Determination of binding affinities using lineshape analysis" in the Methods section, *Supplementary file 2*). These interactions are only moderately affected by the C113S mutation that reduces Pin1 catalytic activity ~40-fold (*Behrsin et al., 2007*). The catalytically deficient C113S Pin1 variant shows a clear bivalent interaction mode with pV5βII (*Figure 4—figure supplements 3 and 4*). Compared to the wild-type (WT) Pin1, C113S has only ~2-fold weaker affinity to pV5βII ($K_d$ value of 3.4 µM, *Supplementary file 2*). The low micromolar $K_d$ values obtained for WT Pin1 and the C113S variant reflect the thermodynamic advantage that the bivalent interaction mode imparts on the interactions with the C-term tail.

We illustrate the thermodynamic effect of bivalency using the binding curves of two representative residues (Ser115 and Phe125) in three distinct system compositions (*Figure 4C*). The affinity of monovalent PPIase-pHMβII interactions does not appreciably depend on whether pTMβII is pre-bound to the WW domain: the corresponding $K_d$ values are 133 µM and 117 µM, respectively (*Figure 4C*, *Supplementary file 2*). Moreover, the $K_d$ values for the monovalent pHMβII-WW and pTMβII-PPIase interactions show little dependence on whether the isolated Pin1 domains or full-length Pin1 were used in the binding experiments (*Figure 4D*). However, when both motifs are presented to full-length Pin1 on the same polypeptide chain, we observe an ~90-fold enhancement in affinity ($K_d$ = 1.5 µM). These general findings hold for the Pin1-pV5α interactions (*Figure 4—figure supplement 5*). In aggregate, these data provide a thermodynamic view of bivalent interactions between Pin1 and the

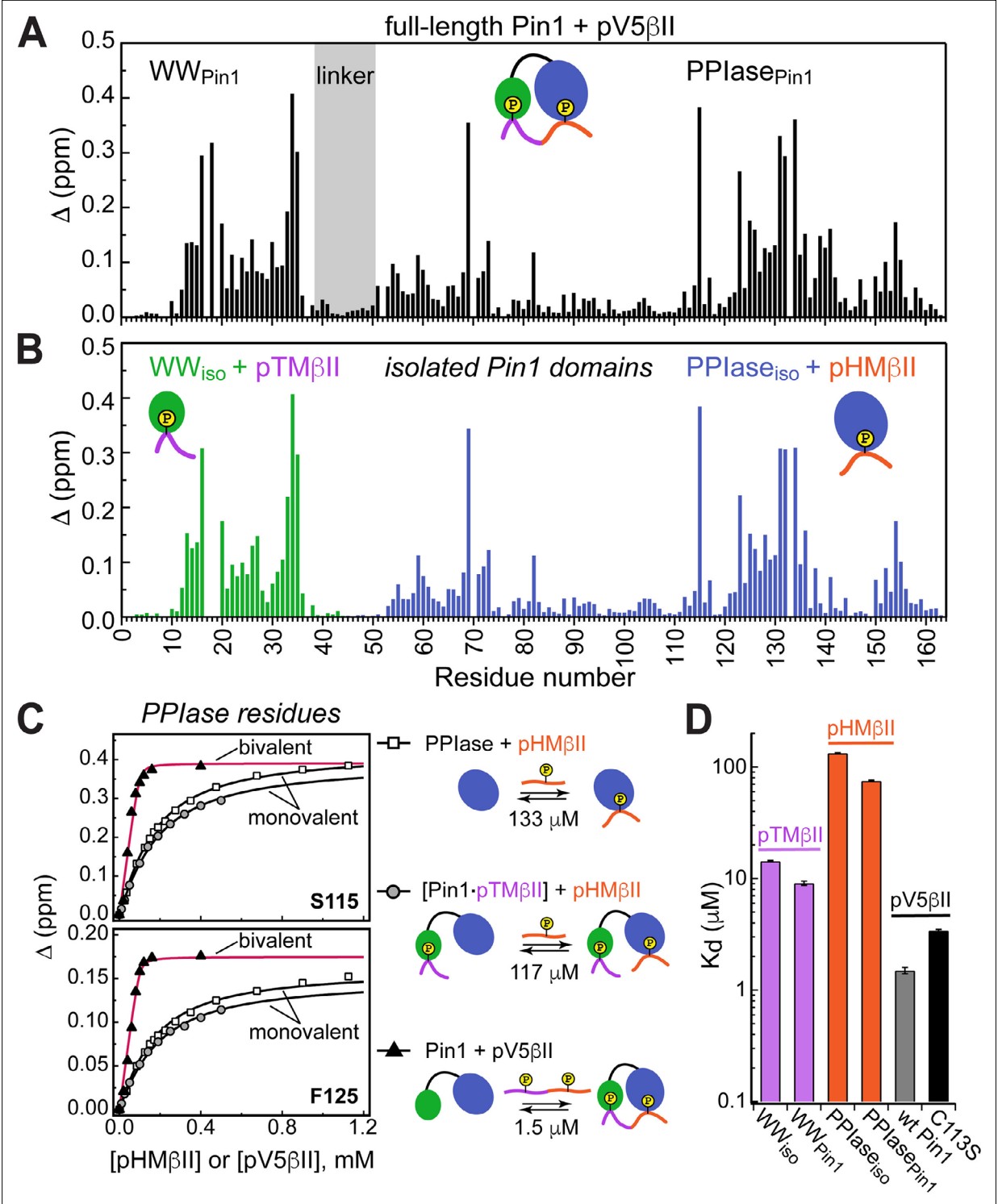

**Figure 4.** Unidirectional bivalent binding mode of the C-terminal PKCβII region to Pin1. (**A, B**) Comparison of the chemical shift perturbation (CSP) plots of Pin1 obtained at maximum concentrations of pV5βII (**A**) and those of isolated domains, WW$_{iso}$ and PPIase$_{iso}$, at maximum concentrations of pTMβII and pHMβII (**B**), respectively. The similarity of CSP patterns in (**A**) and (**B**) indicates that the C-term region of PKCβII binds to Pin1 in a unidirectional bivalent mode. (**C**) The turn motif (TM) and hydrophobic motif (HM) binding sites reside on the WW and PPIase domains, respectively. The protein concentration is 100 μM. Other details are given in **Supplementary file 2**, binding experiment IDs #9, 15, and 18. (**D**) K$_d$ values for the monovalent interactions of the HM and TM with isolated Pin1 domains and full-length Pin1 are contrasted with the K$_d$ value for the bivalent Pin1-pV5βII interactions. ~10-fold enhancement for the pTM binding to the WW domain and ~90-fold enhancement of the pHM binding to the PPIase domain are attributed to

*Figure 4 continued on next page*

*Figure 4 continued*

bivalency. The $K_d$ values used for this plot were obtained in the NMR-detected binding experiments. The $K_d$ value for pV5βII binding to the catalytically deficient C113S Pin1 variant (black bar, 3.4 μM) exceeds the wild-type value by ~2-fold.

The online version of this article includes the following figure supplement(s) for figure 4:

**Figure supplement 1.** NMR-detected binding of the fully phosphorylated C-term regions pV5βII (**A**) and pV5α (**B**) to full-length Pin1.

**Figure supplement 2.** Unidirectional bivalent binding mode of the C-terminal PKCα region to Pin1.

**Figure supplement 3.** The C-term region of PKCβII binds to the catalytically deficient C113S Pin1 variant.

**Figure supplement 4.** Unidirectional bivalent binding mode of the C-terminal PKCβII region to C113S Pin1.

**Figure supplement 5.** Thermodynamic benefits of bivalent of Pin1-C-term PKCα interactions.

C-term PKC tail, and attribute the respective ~10(3)-fold and ~90(60)-fold enhancements of pTMβII(α) and pHMβII(α) binding to Pin1 to bivalency.

## Phosphorylation of the conserved motifs determines the Pin1-C-term interaction mode

The action of phosphatases on the HM and TM generates monophosphorylated and dephosphorylated PKC species that are detected in Pin1 pull-down assays (*Abrahamsen et al., 2012*). To establish how these C-term modifications affect the Pin1 binding mode, we examined the interactions of Pin1 with the α and βII C-term regions where either TM (V5-pTM-HM), HM (V5-TM-pHM), or neither (V5) were phosphorylated. Since clear spectroscopic signatures are available from our data on the PKC motif interactions with full-length Pin1 and its domains (*Figures 1C, 3A, B and 4A ,B*), they can be used to identify the interaction modes of different V5 species. We illustrate this point using two monophosphorylated regions of the βII isoforms bound to full-length Pin1 (*Figure 5A–D*).

The V5-pTM-HM binding to the full-length Pin1 produced a CSP pattern in the WW domain that is identical to that of the isolated pTM (compare *Figures 5A and 1B*) and to that of the pV5βII region (*Figure 5B*, $WW_{Pin1}$ correlation plot). Since pTM binding to $WW_{Pin1}$ does not cause any CSPs in $PPIase_{Pin1}$ (*Figure 1C*), the CSPs in the $PPIase_{Pin1}$ upon V5-pTM-HM binding could only be caused by the interactions with the unphosphorylated HM. The $PPIase_{Pin1}$ chemical shift correlation plot of *Figure 5B* reflects the influence of the HM phosphate group on the backbone chemical shifts of the isomerase domain. In conclusion, phosphorylated TM imposes a unidirectional bivalent binding mode by employing its high-affinity interaction with $WW_{Pin1}$ to direct the unphosphorylated HM to the binding site on $PPIase_{Pin1}$.

The C-term region monophosphorylated at HM (V5-TM-pHM) produced a CSP pattern (*Figure 5C*) that is essentially identical to that of the isolated pHM binding to full-length Pin1. This is illustrated by the chemical shift correlation plots constructed for $WW_{Pin1}$ and $PPIase_{Pin1}$ (*Figure 5D*). Since the unphosphorylated TM does not appreciably interact with $WW_{Pin1}$, it is unable to impose the unidirectional bivalent binding mode, and the phosphorylated HM is free to occupy both the WW- and PPIase-localized binding sites. We conclude that for the V5-TM-pHM region, the mode of interaction with Pin1 is monovalent and lacks directionality.

Using these types of analysis, we identified the binding modes of the V5 regions of α and βII isoforms (*Figure 5E–H*). The chemical shift data (*Figure 5A and C*; *Figure 5—figure supplement 1*) are mapped onto the extended conformation of Pin1 and the binding modes are illustrated with cartoon representations. Of note, the unphosphorylated C-term region interacts with Pin1 weakly. The CSPs are extremely small for the α isoform but are sufficiently large for βII to arrive at a $K_d$ value of ~960 μM (*Supplementary file 2*). Based on our data, we conclude that the phosphorylation status of TM determines the valency of the Pin1-PKC C-term interaction mode.

The $K_d$ values determined for all C-term regions enabled us to estimate the thermodynamic gain associated with the phosphorylation of TM and HM in the context of bivalent interactions (*Figure 5I* and *Supplementary file 2*). The affinity enhancement due to the TM phosphate is 62-fold, which corresponds to $ΔΔG°_{pTM}$ of –2.4 kcal/mol. The affinity enhancement due to the HM phosphate is 14-fold, which corresponds to $ΔΔG°_{pHM}$ of –1.6 kcal/mol. Both $ΔΔG°$ values are within the range reported for the formation of the phosphate-mediated salt bridges in proteins (*Errington and Doig, 2005*). Summing up the contributions from the two phosphate groups produces the overall $ΔΔG°$ of –4.0 kcal/mol, or ~900-fold enhancement of the Pin1 affinity to the phosphorylated C-term of PKCβII.

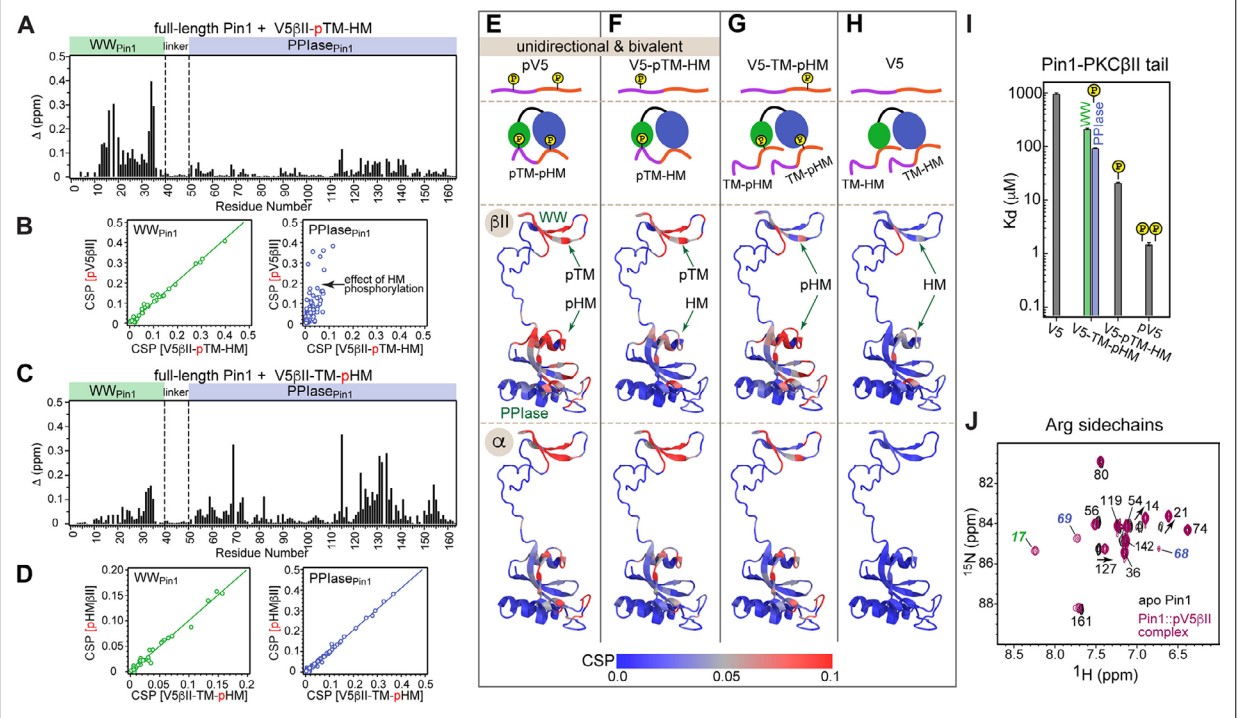

**Figure 5.** Phosphorylation state of the conserved C-term motifs defines the C-term interaction mode with Pin1. (**A**) Chemical shift perturbation (CSP) pattern of full-length Pin1 due to interactions with the monophosphorylated V5βII-pTM-HM region. The concentrations of Pin1 and V5βII-pTM-HM are 100 μM and 0.46 mM, respectively. (**B**) Chemical shift correlation plots between the Pin1::pV5βII and Pin1::V5βII-pTM-HM complexes, plotted separately for the WW (green) and the PPlase (blue) domains. pTM interacts exclusively with the WW domain and thereby imposes a unidirectional bivalent mode. The differences in the PPlase CSPs are due to the differences in the HM phosphorylation state between pV5βII and V5βII-pTM-HM. (**C**) CSP pattern of full-length Pin1 due to interactions with the monophosphorylated V5βII-TM-pHM region. The concentrations of Pin1 and V5βII-TM-pHM are 100 μM and 1.64 mM, respectively. (**D**) Chemical shift correlation plots between the Pin1::pHMβII and Pin1::V5βII-TM-pHM complexes, plotted separately for the WW (green) and the PPlase (blue) domains. High similarity of chemical shifts indicates that pHMβII and V5βII-TM-pHM binding modes are identical. (**E–H**) CSPs of full-length Pin1 due to interactions with the C-term PKC regions mapped onto the extended NMR structure of apo Pin1 (1nmv). The similarity of the Pin1 CSP patterns due to α and βII C-term binding suggest similar Pin1 interaction modes with PKCβII and PKCα isoforms. Phosphorylation of TM imposes a unidirectional bivalent mode irrespective of the phosphorylation state of the HM (**E, F**). Phosphorylation of HM directs HM to its binding sites on the WW and PPlase domains but does not impose a bivalent interaction mode (**G**). (**H**) Pin1 interactions with unphosphorylated C-term are only detectable for the C-term of PKCβII. (**I**) Dissociation constants of the Pin1::C-term(βII) complexes illustrating the enhancement of binding affinity due to phosphorylation. The data for the V5-TM-pHM binding to WW and PPlase are color-coded green and blue, respectively. The protein concentration is 100 μM. Other details are given in **Supplementary file 2**, binding experiment IDs #11, 13, 15, and 16. (**J**) 2D [$^{15}$N-$^{1}$H] heteronuclear single-quantum coherence (HSQC) spectra of the arginine sidechains in apo Pin1 (black) and Pin1::pV5βII complex (maroon). Cross-peaks that are exchanged-broadened in apo Pin1 but reappear upon pV5βII binding belong to the Arg residues in the phosphate binding sites of WW (green) and PPlase (blue). TM, turn motif; HM, hydrophobic motif.

The online version of this article includes the following figure supplement(s) for figure 5:

**Figure supplement 1.** Pin1 chemical shift perturbation (CSP) plots of Pin1 obtained at maximum concentrations of V5βII-TM-HM (**A**), V5α-pTM-HM (**B**), V5α-TM-pHM (**C**), and V5α-TM-HM (**D**).

**Figure supplement 2.** Representative inter-molecular $^{1}$H-$^{1}$H NOEs between the pTM-WW$_{Pin1}$ (**A**) and pHM-PPlase$_{Pin1}$ (**B**).

The phosphate groups occupy the canonical phosphate binding sites of the Pin1 domains. We reach this conclusion by comparing the $^{15}$N-$^{1}$H HSQC spectra of Arg sidechains in the apo and pV5βII-complexed Pin1 (**Figure 5J**). Based on the structural data obtained previously for monovalent substrates (**Verdecia et al., 2000**; **Hu et al., 2017**), Arg17$_{WW}$ and the Arg68-Arg69$_{PPlase}$ motif of the PPlase catalytic loop form salt bridges with the phosphate groups. While these three residues are exchange-broadened in apo Pin1 due to dynamics, their cross-peaks reappear upon pV5βII binding – consistent with their direct interactions with these phosphate groups (**Figure 5J**). To describe the bivalent interaction mode and identify the role of individual PKC and Pin1 residues, we determined the high-resolution structure of the complex.

## Structural basis of the Pin1-PKCβII C-term bivalent recognition mode

While both α and βII isoforms show similar bivalency patterns in their Pin1 interaction modes (*Figure 5A–D*), the 8-fold higher binding affinity of pV5βII to Pin1 informed the choice of the Pin1-pV5βII complex for structural work. Since exhaustive screening of crystallization conditions failed to yield crystals suitable for X-ray diffraction, we pursued a solution NMR-based approach. The NMR structural ensemble of the Pin1-pV5βII complex was calculated with CYANA using $^1$H-$^1$H NOEs (*Figure 5—figure supplement 2A and B*), hydrogen bond, and torsional angle restraints, and refined in explicit solvent using XPLOR-NIH ("NMR structure calculation and refinement" in the Methods section). The Pin1-pV5βII interface is defined by 75 inter-molecular NOEs whose assignment required the use of specifically labeled amino acids incorporated into the peptide substrate; a total of nine different Pin1 complexes were prepared to ensure sufficient data redundancy (*Supplementary file 3*). The overall ensemble has a backbone and all-heavy-atom RMSDs of 0.9 Å and 1.2 Å for the ordered regions, respectively (*Supplementary file 4* and *Figure 5—figure supplement 2C*). Of note, the linker region connecting the two domains retains its flexibility in the complex and confers some variability onto the relative position of the WW and PPIase domains, as is evident from the ensemble representation of *Figure 5—figure supplement 2C*.

The complex exhibits novel structural features that distinguish it from all other structures of Pin1 complexes known to date. These features are highlighted in *Figure 6* using the lowest-energy structure of the ensemble. First, the Pin1 substrate-binding mode is unusual in that pV5βII traverses the entire protein and interacts with both domains in a bivalent arrangement. The terminal pV5βII anchoring points are the two phosphorylated motifs, the pSer660 of the HM that binds to the PPIase domain and the pThr641 of the TM that binds to the WW domain (*Figure 6A*). In addition, the pV5βII region in between the two phosphorylated motifs forms an extensive network of interactions with both WW and PPIase domains (*Figure 6—figure supplements 1 and 2*). The second noteworthy aspect of the structure is the extensive conformational change that Pin1 undergoes upon pV5βII binding. The WW and PPIase domains are brought into proximity primarily via interactions with the N-terminal pV5βII region. This compact conformation of pV5βII-complexed Pin1 is distinct from the 'closed' conformation observed in available crystal structures of monovalent Pin1 complexes (*Figure 6B*). The major differences are: (i) the ~70° rotation of the WW domain relative to the PPIase module; and (ii) the repositioning of the α4 helix and the α4-β2 loop of the PPIase domain to accommodate pV5βII. The linker region connecting the WW and PPIase domains retains its flexibility and is not involved in pV5βII interactions.

## Interface of the pTM anchor and turn regions of PKCβII C-term with Pin1

To facilitate structural analysis of the Pin1-pV5βII interface, we separated pV5βII into four segments: the 'pTM anchor' (639–643, S1), the 'turn' (644–650, S2), the 'groove' (651–655, S3), and the 'pHM anchor' (656–661, S4) (*Figure 6A*, *Video 1*). The five-residue 'pTM anchor' harboring the pThr641-Pro642-Pro643 segment is positioned at the interface between the WW and PPIase domains (*Figure 6C*). Its backbone runs almost parallel to the α1 helix of the PPIase domain. The key interactions are the salt bridge between the Thr641 phosphate and guanidinium groups of Arg17$_{WW}$, and the hydrophobic contacts of the pThr641 methyl group and Pro642 pyrrolidine ring with the sidechains of Trp34$_{WW}$, Asn90$_{PPIase}$, Ile93$_{PPIase}$, and Gln94$_{PPIase}$. In addition, the NHε group of the Trp34$_{WW}$ forms a hydrogen bond with the carbonyl oxygen of pThr641. These interactions are present in 65–100% of the Pin1-pV5βII ensemble structures.

The Pro643-Asp644 segment reorients the backbone such that it runs along the β1 strand of the WW domain. This marks the beginning of the seven-residue 'turn' segment (*Figure 6D*) that completes the inter-domain arm of pV5βII and is responsible for the realignment of the PPIase-WW domain interface. Prior structural work on Pin1 identified residues 137–141$_{PPIase}$, 148–149$_{PPIase}$, and 28–32$_{WW}$ as being involved in the dynamic inter-domain interface (*Born et al., 2019*; *Wilson et al., 2013*). In the Pin1-pV5βII complex, many residues from this subset are now engaged in interactions with 'turn' segment pV5βII (*Figure 6D* and *Figure 6—figure supplement 1*). The turn itself is stabilized by intra-pV5βII hydrogen bonds and salt bridges. Arg649 plays a particularly prominent role in that regard as it participates in both types of interactions. The intra-molecular salt bridge formed by the Arg649 and Glu646 sidechains (present in 75% of the ensemble structures) is unique to the βII

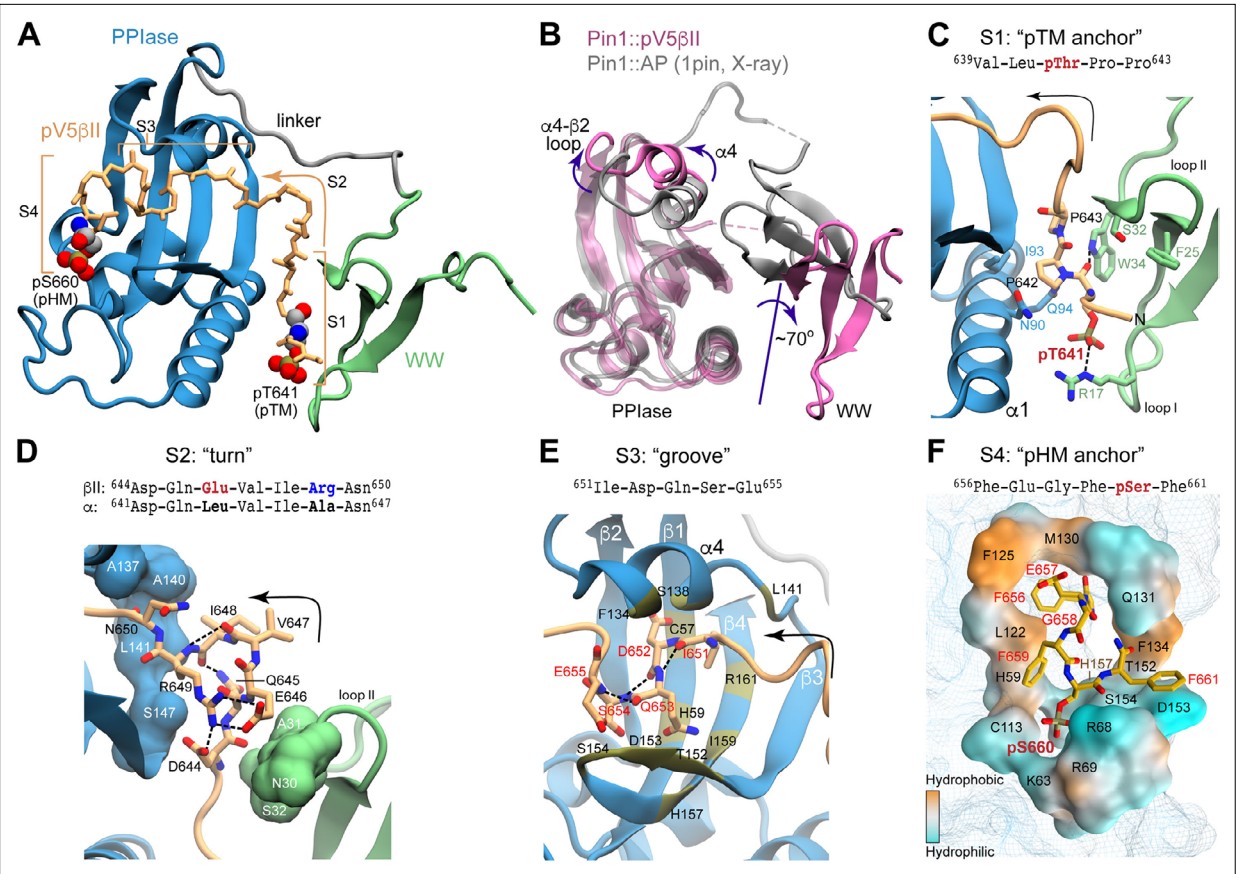

**Figure 6.** Structure of the Pin1::pV5βII complex reveals the bivalent recognition mode. (**A**) The lowest-energy NMR structure showing the pV5βII backbone (tan) forms an extensive binding interface with the WW (green) and PPIase (blue) domains of the full-length Pin1. pV5βII is broken into four segments, S1-S4, to facilitate the structural analysis. The phosphorylated Ser of the hydrophobic motif (HM) and Thr of the turn motif (TM) are shown in van der Waals representation. (**B**) Overlay of the crystal structure of the Pin1-AlaPro complex (1pin) and the NMR structure of the Pin1-pV5βII complex (8SG2, this work), illustrating the 70° rotation of the WW domain, along with the displacement of the α4 helix and the α4-β2 loop. (**C**) The 'pTM anchor' segment is positioned at the interface between WW and PPIase domains. The phosphate group of pThr641 forms a salt bridge with Arg17. (**D**) The 'turn' segment is stabilized by intramolecular hydrogen bonds and is wedged between the WW and PPIase domains. (**E**) The 'groove' segment is threaded between the α4 helix and the β3-β4 hairpin of PPIase. (**F**) The 'pHM anchor' segment occupies the catalytic site of the PPIase domain. Residues forming the site are color-coded according to amphiphilicity. The phosphate group of pSer660 forms salt bridges with the Arg68 and Arg69 residues of the catalytic loop. Hydrogen bonds and salt bridges are shown with black dashed lines.

The online version of this article includes the following figure supplement(s) for figure 6:

**Figure supplement 1.** 2D LigPlot+ diagram of representative Pin1 interactions with residues 639–650 ('pTM anchor' and 'turn') of pV5βII.

**Figure supplement 2.** 2D LigPlot+ diagram of representative Pin1 interactions with residues 651–661 ('groove' and 'pHM anchor' segments) of pV5βII.

**Figure supplement 3.** The C-terminal tail in the structure of the PKCβII catalytic domain (PDB ID 2I0E).

**Figure supplement 4.** The C-terminal part of pV5βII is threaded through the PPIase groove.

**Figure supplement 5.** Comparison of the binding poses between the D-peptide, a potent unnatural peptide inhibitor of Pin1, and the 'pHM anchor' segment of pV5βII.

isoform (*Figure 6—figure supplement 3*) because these Arg and Glu residues are replaced by hydrophobic Leu and Ala/Met residues in other conventional PKC isoforms (*Figure 6D* and *Figure 1A*). The contributions of Arg sidechain-mediated interactions are likely responsible for the higher affinity of pV5βII for Pin1 relative to the affinity of pV5α.

## Interface of the groove and pHM anchor regions of PKCβII C-term with Pin1

The turn is followed by the five-residue pV5βII 'groove' segment that is rich in polar residues and interacts exclusively with the PPIase domain. The groove segment threads through a deep groove in

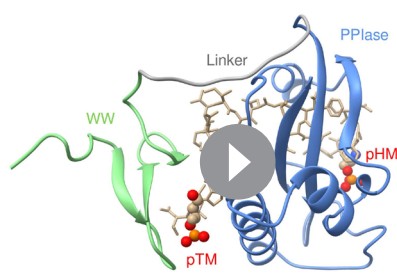

**Video 1.** Visualization of the lowest energy structure of the Pin1::pV5βII NMR ensemble.
https://elifesciences.org/articles/92884/figures#video1

the Pin1 PPIase domain formed by helix α4 and the β-sheet comprised of strands β3, β4, and β1 (*Figure 6E* and *Figure 6—figure supplement 4*). This configuration drives the repositioning of helix α4 and the α4-β2 loop relative to the structures of Pin1 complexed to the monovalent substrate (*Figure 6B*). The Pin1-pV5βII interface involves hydrophobic contacts, present in all 20 ensemble structures, of the only hydrophobic residue of this pV5βII segment (Ile651) with Pin1 residues Leu141$_{PPIase}$, Ile159$_{PPIase}$, and the Arg161$_{PPIase}$ methylenes. Among polar residues, pV5βII residues Ser654 and Glu655 are within H-bonding distance with the three Pin1 residues of the conserved parvulin tetrad (*Wang et al., 2015b*; *Mueller et al., 2011*; *Barman and Hamelberg, 2014*): His59$_{PPIase}$ and His157$_{PPIase}$, and Thr152$_{PPIase}$, respectively (*Figure 6—figure supplement 2*).

The last segment of pV5βII (the 'pHM anchor') emerges from the PPIase groove and occupies the catalytic site. This 'anchor' segment contains the entire PKCβII HM whose key residues (Phe656, Phe659, and pSer660) interact with the amphiphilic environment of the Pin1 catalytic site (*Figure 6F*). Specifically, the sidechains of Phe656 and Phe659 are accommodated by the hydrophobic environment formed by Met130$_{PPIase}$, Phe125$_{PPIase}$, and Leu122$_{PPIase}$. The Phe659 aromatic ring can potentially engage in stacking interactions with His59$_{PPIase}$. The pSer660 phosphate is anchored to the Pin1 catalytic loop via salt bridges with a triad of positively charged residues (Arg68$_{PPIase}$, Arg69$_{PPIase}$, and Lys63$_{PPIase}$). Those interactions, present in all 20 ensemble structures, rigidify the loop as is evident from the reappearance of Arg68$_{PPIase}$ and Arg69$_{PPIase}$ resonances in the NMR spectra upon complex formation (*Figure 5J*). Phe661, the third Phe of the HM, is not involved in any persistent interactions with the Pin1 PPIase module.

We then compared the binding poses of pHM with that of the D-peptide – a potent unnatural peptide inhibitor of Pin1 that binds specifically to the catalytic site of the PPIase domain (*Zhang et al., 2007*; *Figure 6—figure supplement 5A and B*). Structural overlay of the complexes shows that pHM and the D-peptide occupy the same PPIase region, and the positions of the ligand phosphate groups coincide (*Figure 6—figure supplement 5C*). Notable differences include the arrangement of the ligand hydrophobic ring moieties in the catalytic site, and the position of the Gln131$_{PPIase}$ sidechain relative to the ligand. Specifically, the space occupied by the Gln131$_{PPIase}$ sidechain in the Pin1::pV5βII complex is occupied by the Gln5 sidechain of the D-peptide in the Pin1::D-peptide complex (*Figure 6—figure supplement 5C*). Gln131 is the C-terminal residue of the α4 helix that undergoes the most significant rearrangement upon the formation of the bivalent Pin1::pV5βII complex (*Figure 6B*).

## Pin1 null HEK293T cells exhibit elevated steady-state levels of PKCα

The structural data revealed: (i) a bivalent mode of Pin1 interactions with the C-term tail of PKCβII, (ii) that the phosphate groups occupy the canonical binding sites of the Pin1 WW and PPIase domains, and (iii) that the intervening residues can form an extensive network of interactions with the residues of both Pin1 domains. Moreover, the similarities of the NMR CSP patterns between pV5βII and pV5α binding to Pin1 report that the bivalent interaction mode is shared by both the α and βII PKC isoforms. To interrogate our biophysical and structural conclusions regarding Pin1 function in a cellular context, we developed a system for assessing Pin1-mediated PKCα regulation without the contribution of endogenous Pin1 activity in a HEK293T cell model. HEK293T cells were chosen because these cells are efficiently transfected and therefore broadly used in such experiments. PKCα was chosen for these analyses because we were able to detect endogenous PKCα by immunoblotting in HEK293T cells, whereas PKCβII overexpression was required for detection in this cell line.

Previous studies reported Pin1 downregulates PKC levels in serum-starved cells stimulated with PDBu (*Abrahamsen et al., 2012*). Consistent with this general concept, efficient (~80%) siRNA-mediated knockdown of Pin1 expression resulted in an ~45% elevation in steady-state PKCα levels

in cells cultured in the presence of serum. That is, cells stimulated by natural agonists under more physiological conditions than those stimulated by exogenous agonist PDBu under serum starvation (*Figure 7—figure supplement 1A*). This relationship was further corroborated by CRISPR screens that produced Pin1 null HEK293T cell lines ("Isolation of a clonally-derived Pin1 null cell line" in the Methods section; *Figure 7—figure supplement 1B*). Clonally derived cells recovered from the CRISPR challenge exhibited a wide range of Pin1 expression levels, and immunoblotting confirmed an inverse correlation between endogenous Pin1 and steady-state PKCα levels (*Figure 7A*; R=–0.92, p<0.0005). The linearity of this relationship suggests Pin1 is a limiting component in downregulation of PKCα levels.

The CRISPR approach produced three HEK293T clones with little or no detectable Pin1 antigen, and those lines exhibited the highest steady-state levels of PKCα. Clone 3 represented a particularly attractive candidate for a Pin1 null cell line as it was devoid of detectable Pin1 antigen. This was confirmed by DNA sequencing. HEK293T cells carry four copies of the Pin1 gene (*Lin et al., 2014*), and DNA sequence analyses indicated clone 3 harbored three frameshift alleles. All three alleles altered the targeted exon 2 that encodes the Pin1 PPIase domain and included: (i) a 5 bp deletion that interrupts the Pin1 amino acid sequence after Thr79, (ii) an 8 bp insertion that interrupted the Pin1 sequence after Arg80, and (iii) a 10 bp deletion that disrupts the Pin1 amino acid sequence after Lys77 (*Figure 7—figure supplement 1C*). The resulting translation products were prematurely terminated after addition of another 22, 10, and 7 residues, respectively. Although PKCα steady-state levels were upregulated approximately 2-fold in clone 3 relative to parental WT HEK293T cells under both serum-free and serum-replete conditions (*Figure 7B*; 2.3 ± 0.2- and 2.3 ± 0.3-fold, respectively), the Pin1 null cells were not phenotypically perturbed. Their dimensions (diameter, surface area), viabilities, and proliferation rates were indistinguishable from those of the parental cells (*Figure 7—figure supplement 1D–G*). Thus, the elevation in steady-state PKCα levels identified a baseline Pin1 loss-of-function phenotype.

## Differential effects of Pin1 mutant expression on PKCα homeostasis in cultured cells

To assess the effects of Pin1 mutants on steady-state PKCα expression without the contribution of endogenous Pin1 activity, stable transgenic cell lines individually expressing epitope-tagged versions of WT Pin1; substrate-binding mutants: $Pin1^{W34A}$, $Pin1^{R68A,R69A}$, $Pin1^{W34A,R68A,R69A}$; or the catalytic-deficient $Pin1^{C113S}$ were derived from clone 3 cells. As shown in *Figure 7C* (top left panel), WT Pin1 and all three of the Pin1 substrate-binding mutants were overexpressed some 4- to 10-fold relative to endogenous Pin1 levels in these stable transgenic lines. Assessment of steady-state PKCα levels in the reconstituted cell lines showed that WT Pin1 expression significantly reduced PKCα steady-state levels relative to the Pin1 null condition (*Figure 7C*, right panel). By contrast, reconstituted expression of neither $Pin1^{W34A}$, $Pin1^{R68A,R69A}$, nor of $Pin1^{W34A,R68A,R69A}$ restored Pin1 function (*Figure 7C*, right panel). As expected, the differences in the triple mutant data in comparison to the Pin1 null mutant data were not statistically significant (p=0.07). However, even though the data did not reach a threshold of statistical significance, PKCα levels in the $Pin1^{W34A,R68A,R69A}$-expressing cells were consistently higher than those in Pin1 null cells. One speculation is the triple mutant imposes dominant negative effects on some limiting factor in PKCα degradation that are revealed in the Pin1 null background. More work is required to resolve this issue.

Interestingly, we were unable to produce stable cell lines that overexpressed the $Pin1^{C113S}$ 'catalytic-dead' mutant to the same levels achieved for WT Pin1 and the three substrate-binding mutants. Reconstituted $Pin1^{C113S}$ expression was consistently elevated only some 3-fold relative to endogenous Pin1 levels (*Figure 7C*, bottom left panel). Yet, this relatively modest level of $Pin1^{C113S}$ expression was as effective as WT Pin1 expression in reducing steady-state PKCα levels in otherwise Pin1 null HEK293T cells (*Figure 7C*, right panel). These collective results support a physiologically relevant mechanism for Pin1-mediated regulation of PKCα that requires a bivalent interaction mode of Pin1 with PKCα, but operates independently of the Pin1 prolyl-isomerase catalytic activity.

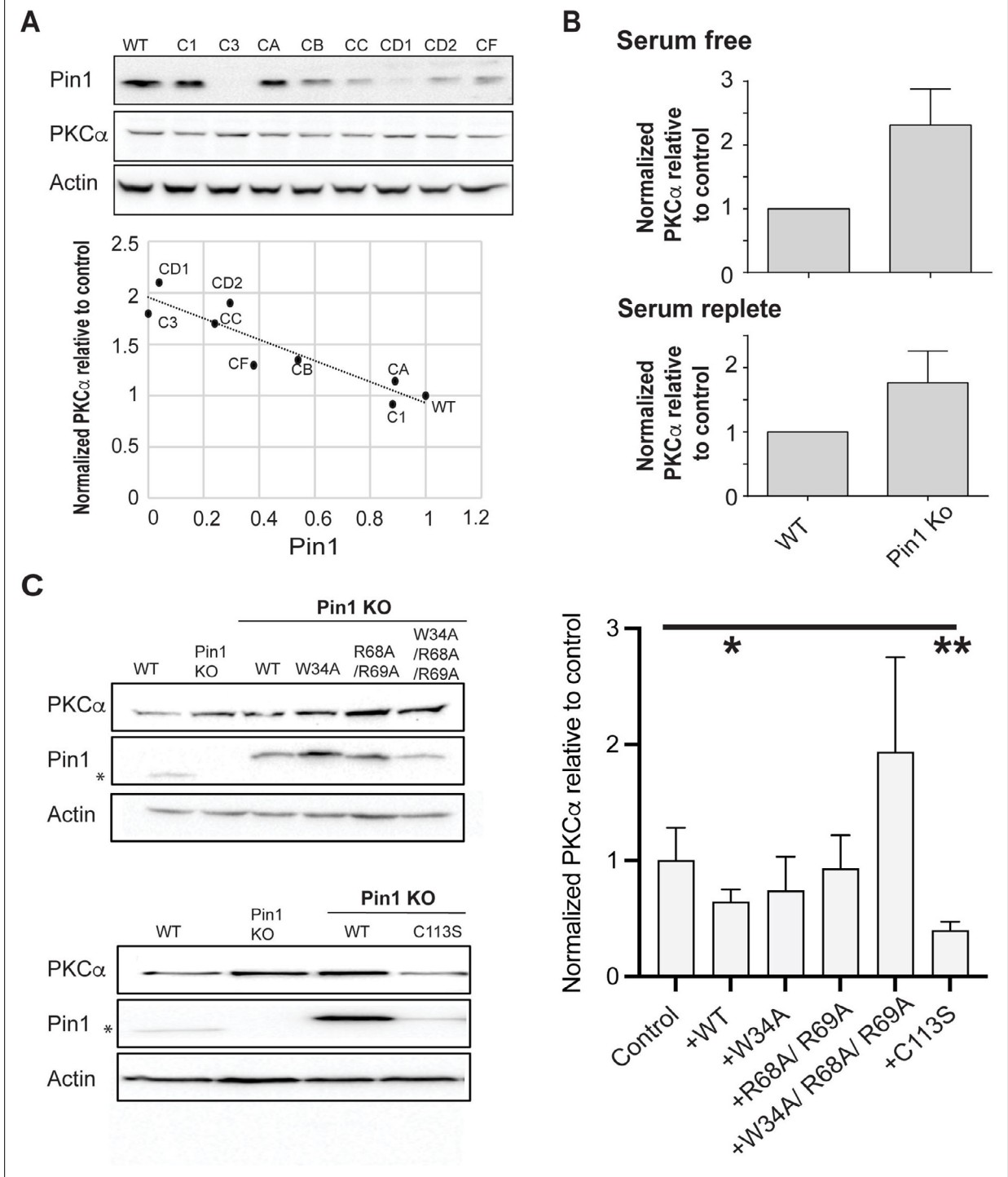

**Figure 7.** Regulation of PKCα homeostasis by Pin1 in HEK293T cells. (**A**) PKCα protein levels at steady-state are inversely proportional to Pin1 levels. HEK293T cells were transfected with CRISPR/Cas9 plasmids encoding Pin1 guide RNAs and clonal lines were generated. Lysates of the clonally derived cell lines were resolved by SDS-PAGE, transferred to nitrocellulose, and immunoblots developed to visualize PKCα, Pin1, and actin. Blot profiles are shown at top. Bottom panel relates steady-state PKCα protein levels to Pin1 steady-state levels. Actin was used to normalize the PKCα and Pin1 profiles for each cell line and the PKCα/actin ratio was set as 1.0 for parental wild-type (WT) Pin1 cells. Clone 3 (C3) expressed no detectable Pin1 antigen and was selected for further study. (**B**) Pin1 null HEK293T cells exhibit elevated steady-state PKCα protein levels when cells are incubated under serum-free (upper panel; 2.3 ± 0.2; n=7; p<0.0001, two-tailed t-test) and serum-replete conditions (lower panel; 2.3 ± 0.3; n=12; p<0.0002, two-tailed t-test). Actin was used to normalize the PKCα and Pin1 immunoblot profiles for each cell line and the PKCα/actin ratios were set as 1.0 for parental WT Pin1 cells. (**C**) PKCα regulation by Pin1 derivatives with defined biochemical defects. Left panels: At top are shown representative PKCα,

*Figure 7 continued on next page*

*Figure 7 continued*

Pin1, and actin immunoblot profiles for WT HEK293T cells, Pin1 KO cells (C3), and Pin1 KO cells stably expressing the indicated mutant Pin1 proteins defective in PKCα binding. At bottom are shown representative PKCα. Pin1 and actin immunoblot profiles for WT HEK293T cells, Pin1 KO cells (C3), and Pin1 KO cells stably expressing the 'catalytic-dead' Pin1^C113S mutant. In both panels the asterisk denotes endogenous Pin1 as the ectopically expressed Pin1 proteins are 3.6 kDa larger in molecular mass due to the myc and DDK epitopes with which these are tagged at their C-termini (Pin1-TRTRPL<u>EQKLISEEDL</u>AANDILDYK<u>DDDDK</u>V). Right panel: Quantification of PKCα steady-state levels in Pin1 null cells (control) and Pin1 null cells reconstituted for expression of the indicated mutant Pin1 proteins as indicated at bottom. For quantification, actin was used to normalize the PKCα and Pin1 profiles for each cell line and the PKCα/actin ratios were set as 1.0 for control Pin1 KO cells. Data represent the averages of five independent biological replicates ± standard deviation. Values were related to Pin1 KO control using an unpaired two-tailed t-test (* p<0.05; **p<0.01). The uncropped immunoblots are provided as source data.

The online version of this article includes the following source data and figure supplement(s) for figure 7:

**Source data 1.** Original uncropped immunoblots for data in *Figure 7A*.

**Source data 2.** Original uncropped immunoblots for data in *Figure 7C*.

**Figure supplement 1.** PKCα levels in cells with reduced Pin1 function.

**Figure supplement 1—source data 1.** Original uncropped immunoblots for data in *Figure 7—figure supplement 1A*.

## Discussion

How Pin1 regulates the activities of its many cellular substrates is of intense interest in contemporary biomedical science. This interest follows not only from the fact that the many targets of Pin1 regulation themselves play important cellular functions, but also that dysregulation of the Pin1 activity underlies the principal basis for multiple pathological disorders in humans. Pin1 has been the focus of many previous studies and it is generally accepted that prolyl *cis-trans* isomerase activity is its obligate functional feature. Yet, our understanding of how Pin1 engages its various substrates and regulates their activities remains incomplete. In this work, we provide a comprehensive biophysical, structural, and cell biological description for how Pin1 recognizes/binds its PKCα and PKCβII substrates, and report the key features required for Pin1-mediated regulation of PKC degradation. Contrary to current dogma, we demonstrate a non-catalytic role for Pin1 in the regulation of PKC stability. We further show the underlying mechanism involves a bivalent interaction mode that has not been previously observed in any reported structures of Pin1 complexes. These discoveries not only expand the potential mechanisms by which Pin1 regulates the activities of its client substrates, but also hold interesting implications for the design of therapeutically effective Pin1 inhibitors.

### Proline at position +1 disqualifies Ser/Thr-Pro Pin1 binding motifs as isomerizable substrates

Previous experimental data suggested that Pin1-binding motifs with Pro at position +1 are disfavored substrates because of their failure to bind the catalytic PPIase domain of Pin1 (*Innes et al., 2013*). Our results provide direct experimental support for that conclusion, as both pTMα and pTMβII motifs bind with high affinities to the Pin1 WW domain but lack high-affinity interactions with the PPIase domain. In addition, EXSY experiments clearly show that Pin1 is unable to catalyze the *cis-trans* isomerization of the pThr-Pro motifs of PKCα and PKCβII. The unsuitability of these motifs as Pin1 substrates is due solely to the presence of Pro at the +1 position as evidenced by our demonstrations that replacement of this Pro residue restores their activities as substrates for isomerization by Pin1. Of note, the inability of Pin1 to catalyze *cis-trans* isomerization was previously observed for the pT^231PP site in the Tau protein (*Smet et al., 2005*; *Eichner et al., 2016*). Interestingly, this observation prompted the speculation that Tau is not a genuine target for regulation by Pin1 – a speculation that our data indicate requires reconsideration (see below).

Why is Pro at position +1 incompatible with Pin1-catalyzed *cis-trans* isomerization of what is otherwise a signature substrate motif? Computational studies predict unfavorable overall binding free energies for Pin1 engagement with the transition state and the *cis*-conformations of the pTPP substrates (*Momin et al., 2018*). A definitive answer as to why Pro(+1)-containing motifs are incompatible with Pin1 catalysis remains elusive as there is currently no consensus on the precise catalytic mechanism of Pin1. The most recent 'twisted-amide' catalysis model (*Mercedes-Camacho et al., 2013*) envisions a transition state where distortion of the substrate is stabilized by the H-bond between the C=O of the substrate Pro at position (0) and the backbone -NH of Pin1 residue Gln131. As formation and

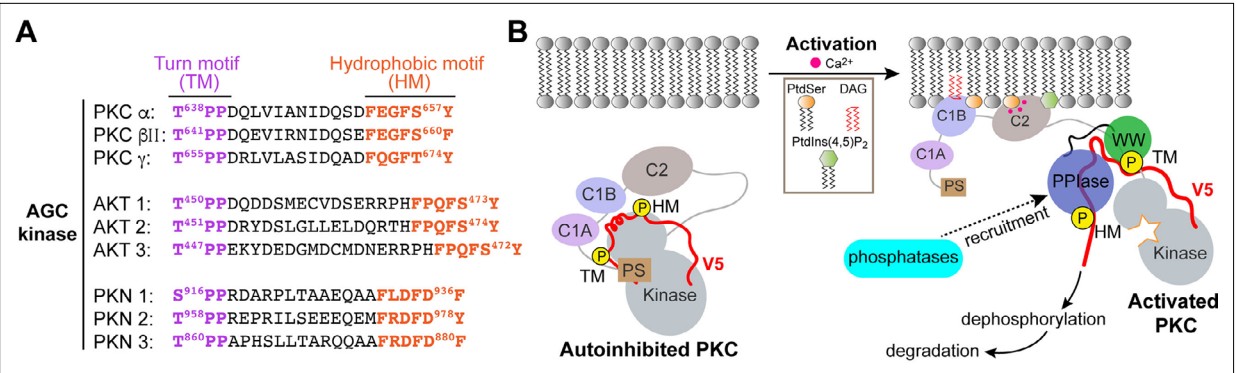

**Figure 8.** Non-catalytic role for Pin1. (**A**) Non-isomerizable pSer/Thr-Pro-Pro turn motifs separated by <25 residues from the hydrophobic motifs are present in the C-term tails of other AGC kinases, such as AKT and PKN. (**B**) A possible model for Pin1-mediated downregulation of PKCα and βII isoforms. In the compact autoinhibited state, the pseudosubstrate (PS) blocks the catalytic site of the kinase and the C-terminal tail is not accessible. PKC activation involves $Ca^{2+}$-dependent recruitment to the membranes where the regulatory domains, C1 through C2, bind diacylglycerol (DAG), phosphatidylserine (PtdSer), and phosphatidylinositol 4,5-bisphosphate (PtdIns(4,5)$P_2$), and thereby trigger the release of autoinhibitory interactions. The activated open PKC conformation exposes the C-terminal V5 domain that is then engaged by Pin1 via bivalent binding to the phosphorylated turn motif (TM) and hydrophobic motif (HM). Pin1 might facilitate the recruitment of phosphatases to PKC by stabilizing the PKC open form. This promotes dephosphorylation of the C-term motifs and subsequent ubiquitination and degradation of the kinase.

stabilization of the transition state relies on a dynamic network of H-bonds involving conserved residues in the Pin1 active site (*Mueller et al., 2011*; *Barman and Hamelberg, 2014*; *Xu et al., 2014*), Pro at position +1 might impose geometric constraints on the substrate where formation or stabilization of the amide twist is disfavored.

The demonstration that Pin1 S/TPP binding motifs are not isomerizable has interesting implications. A survey of AGC kinases reveals that these motifs are surprisingly common in this important class of enzymes. Moreover, as is the case for the PKC kinases, S/TPP motifs are present in the TM regions for both Akt and PKN kinases (*Figure 8A*). These arrangements suggest that Pin1 binding to non-isomerizable substrates through its WW domain defines an unappreciated, yet broadly deployed, regulatory strategy that operates independently of Pin1 *cis-trans* isomerase activity. This point is discussed in further detail below.

## The Pin1-PKC interface is described by a novel bivalent interaction mode

Our NMR structure and NMR-detected binding experiments demonstrate that Pin1 interacts with the C-terminal tails of PKCα and PKCβII in a bivalent mode. To our knowledge, this is the first structure of the full-length Pin1 complexed to the substrate that engages both Pin1 domains. The Pin1 WW domain binds the TM, a canonical Pin1 pThr-Pro recognition motif, whereas the Pin1 PPIase domain engages the HM, a non-canonical substrate that lacks Pro after the pSer residue. The specificity of TM binding exhibited by the WW domain imposes a unidirectionality to the Pin1-PKC interaction by directing the HM to the PPIase domain. This bivalent binding mode results in Pin1 adopting a mostly compact conformation, with the 'pTM anchor' positioned at the interface between two Pin1 domains. Based on our structure, the linker length separating two phosphorylated Ser/Thr in bivalent Pin1 substrates is projected to be an important factor. In the PKCβII V5 domain, the phosphorylated TM and HM sites are separated by 18 amino acids. This length allows the simultaneous binding of the two sites by Pin1 without causing conformational strain on Pin1 itself. It is of interest to consider these results in light of what has been observed in the interaction of the yeast Pin1 homolog Ess1 with the RNA polymerase II C-terminal tail (*Namitz et al., 2021*). The Ess1 linker that joins the WW and PPase domains is a rather rigid element, and CTD repeats exceeding 27 amino acid residues are required to separate the two phosphorylated Ess1-binding sites for bivalent interactions (*Namitz et al., 2021*). By contrast, the cognate Pin1 linker is unstructured, and its flexibility might afford Pin1 the potential to interact with a broader range of substrates. Moreover, the retention of linker flexibility in the Pin1::pV5βII complex suggests that the complexed Pin1 can potentially adopt 'extended' states that would not be readily detectable by the ensemble-averaged methods such as solution NMR.

Bivalent interactions afford significant advantages to biologically relevant protein-protein interactions as these are capable of increasing binding affinity and specificity, and inducing the appropriate conformational rearrangements in binding partners (*Mammen et al., 1998*; *Erlendsson and Teilum, 2020*). It is in this manner that the two substrate-binding domains of Pin1 govern its ability to engage in bivalent binding interactions that enhance the binding affinity and specificity to multi-phosphorylated client proteins (*Daum et al., 2007*; *Rogals et al., 2016*). Several Pin1 substrates that contain neighboring canonical pSer/Thr-Pro motifs in their unstructured regions have been characterized: IRAK1 kinase (*Rogals et al., 2016*), Cdc25c (*Innes et al., 2013*), Tau (*Eichner et al., 2016*), and STAT3 (*Lufei et al., 2007*). Of those, the IRAK1-derived peptide was the only reported case of bivalent binding to Pin1 based on the NMR CSP data analysis, and this interaction involved two isomerizable pSer/Thr-Pro motifs (*Rogals et al., 2016*). Our Pin1::pV5βII complex structure represents the first natural bivalent substrate-bound Pin1 structure. The two phosphate groups occupy the substrate-binding site of the WW and PPIase domains, and the intervening residues stabilize the domain interface and bring the two domains into proximity. The linker that connects the Pin1 WW and PPIase domains maintains its flexibility and does not participate in interactions with the PKCβII C-terminal tail. The structure of this complex provides a guide for interpreting the potential binding modes of other Pin1 substrates that contain multiple pSer/Thr-Pro and non-canonical Pin1 motifs. For example, cyclin E contains both canonical $pThr^{380}$-$Pro^{381}$ and non-canonical $pSer^{384}$-$Gly^{385}$ motifs (*Yeh et al., 2006*). The $pSer^{384}$-$Gly^{385}$ motif serves as an important determinant of cyclin E interactions with Pin1 and its Pin1-mediated downregulation.

## Pin1 regulates PKC levels via a non-canonical non-catalytic mechanism

With regard to Pin1-mediated regulation of PKC, we note the linear relationship between Pin1 and PKCα steady-state levels up to the point of physiological levels of Pin1 expression – as documented in analyses of the clones derived from the CRISPR screen (*Figure 7A*). However, additional elevations in Pin1 expression did not result in further reductions in PKCα steady-state levels as evidenced by the Pin1 reconstitution experiments. Our interpretation of the data is that physiological expression of Pin1 in HEK293T cells is normally the limiting factor in the stimulated PKCα degradation pathway. When Pin1 is expressed in excess of physiological levels however, some other component of the PKCα degradation pathway becomes limiting in those contexts. It is posited that Pin1 isomerase activity promotes ubiquitination and subsequent degradation of conventional PKC enzymes (*Abrahamsen et al., 2012*). This proposal rests on the observation that challenge of stimulated cells with a small molecule inhibitor of Pin1 abolishes agonist-induced ubiquitination of PKC. Interpretation of those data is neither ironclad, nor are those results inconsistent with our conclusions. It remains to be established that the inhibitor does not compromise the Pin1-PKC interaction. As described above, our collective biochemical and biophysical data suggest Pin1 downregulates PKC via a mechanism independent of its prolyl *cis-trans* isomerization activity. Pin1 reconstitution experiments provide direct support for this concept as expression of the 'catalytic-dead' $Pin1^{C113S}$ mutant rescues the Pin1 null condition in a HEK293T cell model.

How might Pin1 operate via a non-catalytic mechanism to downregulate PKC activity? Upon activation, the membrane-associated conformation of PKC is sensitive to dephosphorylation – first at the HM site by PHLPP and subsequently at the TM site and activation loop by PP2A (*Gao et al., 2008*; *Baffi et al., 2019*; *Hansra et al., 1996*; *Baffi and Newton, 2022*). One possibility is that Pin1 discharges a scaffolding or substrate chaperoning function where it aids (directly or indirectly) in the recruitment of protein phosphatases to the Pin1-PKC complex. Indeed, interactions between Pin1 and PP2A have been detected in pulmonary eosinophils (*Shen et al., 2008*). Such an activity, when coupled to a substrate chaperoning function, might 'organize' and stimulate ordered dephosphorylation of the PKC C-terminus with subsequent degradation of the kinase. A recruitment mechanism of this nature is attractive in that it provides a means for channeling activated PKC to specific protein phosphatase(s) in a temporally and spatially appropriate manner (*Figure 8B*). Alternatively, sequestration of the HM site by Pin1 might prevent the formation of the autoinhibitory state (*Yang et al., 2018*; *Antal et al., 2015a*) and thereby 'trap' PKC in an open 'activated' conformation. This activated conformer would be particularly susceptible to the action of phosphatases that trigger its ultimate degradation. The significance of phosphorylated HM in maintaining the stability of AGC kinases is well documented (*Baffi et al., 2019*; *Baffi and Newton, 2022*; *Mora et al., 2004*).

It is tempting to speculate that non-catalytic functions for peptidyl-prolyl isomerases are more broadly represented in this group of enzymes. Pin1 and *Escherichia coli* trigger factor (TF) are both peptidyl-prolyl *cis-trans* isomerases – although Pin1 is a member of the Parvulin family of peptidyl-prolyl isomerases whereas TF is a member of the FK506-binding protein class. The enzymatic activity of TF is dispensable for its function as a protein folding chaperone (*Kramer et al., 2004*). A TF[F198A] mutant competent for client protein binding, but defective in peptidyl-prolyl *cis-trans* isomerization activity, remains active as a folding-promoting chaperone in vivo (*Kramer et al., 2004*). Regardless, our data indicate one can no longer confidently infer whether a protein is a target of Pin1 regulation solely based on whether or not the putative substrate presents isomerizable pSer/Thr-Pro motifs.

## Implications for therapeutic interventions targeting Pin1

The oncogenic properties associated with dysregulation of Pin1 activities identify the isomerase as an attractive target for the development of new therapeutic approaches for cancer treatment (*Zhou and Lu, 2016*; *Wu et al., 2022*; *Chuang et al., 2021*). Unfortunately, although numerous small molecule Pin1 inhibitors have been identified (*Moore and Potter, 2013*; *Yu et al., 2020*), these have been plagued by serious off-target toxicities. Resolution of this issue presents a significant obstacle to productive development of the Pin1-targeted drug pipeline (*Moore and Potter, 2013*; *Fila et al., 2008*; *Nam et al., 2001*). In that regard, the screens used to identify Pin1 inhibitors typically rely on readouts of catalytic activity. The data we report herein suggest alternative strategies. That is, screens for ligands that target the bivalent interaction mode rather than the Pin1 catalytic activity. Such strategies hold the potential for enhancing inhibitor specificity and affinity and thereby reducing the toxicities associated with off-target effects. The structure of the Pin1::pV5βII complex now offers a precise template for guiding the design and development of bivalent inhibitors that simultaneously bind to both the Pin1 WW and PPIase domains.

## Methods

### Protein expression and purification

A total of three Pin1 protein constructs were used in this study. The following genes with a codon-optimized DNA sequence were cloned into a pET-SUMO vector (Invitrogen): full-length Pin1 (residues 1–163), the WW domain (residues 1–50), and the PPIase domain (residues 50–163). Pin1 and isolated domains were heterologously expressed in *E. coli* BL21(DE3) and BL21(DE3)pLysS strains, respectively. For the natural abundance protein preparation, cells were grown in Luria broth (LB) to an $OD_{600}$ of 0.6 prior to induction of protein expression with 0.5 mM IPTG. The cells were grown for additional 4–5 hr at 37°C. For the expression of isotopically enriched proteins, we used an LB to M9 minimal media resuspension method (*Marley et al., 2001*). To generate uniformly $^{15}$N-enriched (U-[$^{15}$N]) or $^{15}$N, $^{13}$C-enriched (U-[$^{15}$N,$^{13}$C]) protein samples, the M9 media contained either 1 g/l of $^{15}$NH$_4$Cl and 3 g/l of natural-abundance D-glucose, or 1 g/l of $^{15}$NH$_4$Cl and 3 g/l of $^{13}$C-D-glucose, respectively. The expression of isotopically enriched Pin1 in BL21(DE3) cells was induced for 15 hr at 15°C. The expression of isotopically enriched individual domains in BL21(DE3) pLysS strain was induced for 5 hr at 37°C.

The cells were harvested by centrifugation (4000 rpm, 30 min) at 4°C. Cell pellets were resuspended in a buffer containing 20 mM Tris-HCl (pH 7.5), 0.5 M NaCl, 5 mM imidazole, and 10 mM β-mercaptoethanol. The 6×His-tagged SUMO fusion proteins were purified using a HisTrap HP Ni$^{2+}$ affinity column (GE Healthcare Life Sciences). The fractions containing fusion protein were pooled and exchanged into a SUMO protease cleavage buffer (20 mM Tris-HCl at pH 8.0, 0.15 M NaCl) using a HiPrep 26/10 column (GE Healthcare Life Sciences). The cleavage reaction was initiated by adding 6×His-tagged SUMO protease to a final concentration of ~8 µg/ml to the protein solution. After 30 min at room temperature, the reaction mixture was loaded onto to the second HisTrap HP Ni$^{2+}$ affinity column to purify the desired protein from the 6×His-tagged SUMO and 6×His-tagged SUMO protease.

Purified proteins (Pin1, WW, or PPIase) were buffer-exchanged into an NMR buffer using either 5 kDa or 3 kDa MWCO centrifugal concentrators (Vivaspin, Sartorius). The NMR buffer contained 10 mM d$_4$-imidazole at pH 6.6, 100 mM KCl, 1 mM TCEP, 8% D$_2$O, and 0.02% NaN$_3$. The purity was assessed with SDS-PAGE conducted on samples with serial dilutions. Protein concentrations were determined by measuring the absorbance at 280 nm and using the following extinction coefficients:

20970 $M^{-1}$ $cm^{-1}$ (Pin1), 13980 $M^{-1}cm^{-1}$ (WW), and 6990 $M^{-1}$ $cm^{-1}$ (PPIase). The molecular masses of purified proteins were determined by MALDI-TOF mass spectrometry.

## Synthesis and purification of the C-terminal PKC peptides

All 18 peptides used in this study were derived from the C-terminal V5 regions of either βII, α, or βI PKC isoenzymes (*Homo sapiens*) and synthesized commercially (*Supplementary file 1*). The modifications included acetylation and amidation at the N- and C-termini, respectively, to avoid the influence of positive and negative charges terminal charges on binding and catalysis. All natural abundance peptides were purchased as crude mixtures and subsequently HPLC-purified on a C18 column (Waters). The buffers used for purification were 5 mM $NH_4HCO_3$ in 100% water (buffer A) and 5 mM $NH_4HCO_3$ in 82% acetonitrile/18% $H_2O$ (buffer B). A linear gradient of buffer B ranging from 0% to 20–40%, depending on the peptide, was applied during the elution step. The molecular weight and purity of the peptides were verified using ESI and MALDI-TOF mass spectrometry (*Figure 1—figure supplement 5*). Isotopically enriched peptides, #14 and #15 in *Supplementary file 1*, were purchased in already purified form and used as is. For the binding experiments (vide infra), stock solutions of peptides were prepared by dissolving the lyophilized powders in $ddH_2O$. The pH was adjusted to 6.6 either with $NH_4OH$ or HCl. The peptide concentration was determined using the absorbance at 205 nm; for the phosphorylated peptides, phosphate assay was additionally used (*Fogg and Wilkinson, 1958*).

## Quantitative analysis of Pin1 interactions with the C-terminal PKC regions

NMR-detected binding experiments were conducted by adding aliquots of concentrated peptide stock solutions to the [U-$^{15}$N] protein (Pin1, WW, or PPIase) samples, and recording [$^{15}$N,$^{1}$H] HSQC spectra for each peptide concentration point. The spectra were collected at 25°C on Bruker Avance III spectrometers operating at the $^{1}$H Larmor frequencies of 600 MHz or 500 MHz. The concentration of protein samples was between 90 μM and 100 μM. A total of 19 binding experiments were conducted; the parameters are summarized in *Supplementary file 2*. The binding data were analyzed in three ways, as described in the following subsections.

### CSP analysis

The CSP analysis involved the calculation of the N-$H_N$ cross-peak displacement between the two proteins states, apo Pin1 and Pin1 in the presence of peptide at some specific concentration used in the binding experiments. The combined CSP value, Δ, was calculated in a residue-specific manner using the following equation:

$$\Delta = \sqrt{(\Delta\,\delta_H)^2 + (0.152 \times \Delta\,\delta_N)^2} \tag{1}$$

### Construction and analysis of the chemical shift-based binding curves

The residues for analysis were selected according to the following criteria: (i) their CSP Δ exceeds the mean; (ii) they fall into the fast-exchange regime; and (iii) their cross-peaks are resolved. These sets of residues (ranging in numbers from 14 to 44) are listed in *Supplementary file 2* for each Pin1/WW/PPIase-peptide ligand pair. A total of 459 binding curves were constructed by plotting Δ values against total peptide concentration. Quantitative analysis was carried out by globally fitting the binding curves for each Pin1/WW/PPIase-peptide pair. Two binding models were used: single-site binding and two independent sites binding where appropriate. The experimental data fit well with the chosen models, indicating that Δ values of the selected residues report directly on binding rather than indirect allosteric/dynamic effects. All $K_d$ values reported in this manuscript were obtained from the global fits.

### Single-site binding model

In this model, the following equation was used for fitting the data is *Auguin et al., 2004*; *Phizicky and Fields, 1995*:

$$\Delta = \Delta_{max} \frac{P_0 + L_0 + K_d - \sqrt{(P_0 + L_0 + K_d)^2 - 4P_0L_0}}{2P_0} \tag{2}$$

where $K_d$ is the dissociation constant, $\Delta_{max}$ is the chemical shift changes at complete saturation, and $P_0$ and $L_0$ are total protein and ligand concentrations, respectively.

## Two-site binding model (independent sites)

According to our NMR data, the binding sites of the pHM on the Pin1 domains are independent, i.e., the binding to second site does not depend on whether or not first site is populated, and vice versa. The formalism outlined below is adapted from *Wang, 1990*. There are two binding equilibria that describe the process:

$$P + L \underset{\phantom{K_{d,W}}}{\overset{K_{d,W}}{\rightleftharpoons}} P_W L \tag{3}$$

$$P_W L + L \underset{\phantom{K_{d,I}}}{\overset{K_{d,I}}{\rightleftharpoons}} PL_2 \tag{4}$$

Definitions: P, full-length Pin1; L, ligand, in this case pHM; $P_W$, Pin1 bound to L through the WW domain; $P_I$, Pin1 bound to L through the PPIase domain; $K_{d,I}$, dissociation constant for L binding to the PPIase domain; $K_{d,W}$, dissociation constant for L binding to the WW domain; $A_{W,i}$ chemical shift difference between the apo and L-bound Pin1 for the ith residue in the WW domain; $A_{I,i}$ max chemical shift difference between the apo and L-bound Pin1 for the ith residue in the PPIase domain; $f_{L,W}$, fractional population of L-bound sites on the WW domain of Pin1; $f_{L,I}$ fractional population of L-bound sites on the PPIase domain of Pin1.

The chemical shift change due to ligand binding will be proportional to the fraction of the protein complexed to ligand L through a particular domain. In our case, we can equate this to the fractional population of ligand-bound sites of the WW and PPIase domains. These fractions can be expressed in terms of corresponding $K_d$ values and free ligand concentration:

$$f_{L,W} = \frac{[L]}{K_{d,W} + [L]} \tag{5}$$

$$f_{L,I} = \frac{[L]}{K_{d,I} + [L]} \tag{6}$$

The expressions for the residue-specific chemical shift changes therefore become:

$$_{W,i} = A_{W,i} f_{L,W} = \frac{A_{W,i}[L]}{K_{d,W} + [L]} \tag{7}$$

$$_{I,i} = A_{I,i} f_{L,I} = \frac{A_{I,i}[L]}{K_{d,I} + [L]} \tag{8}$$

*Wang, 1990*, provided the expression for [L] in terms of $L_0$ and $P_0$:

$$[L] = -\frac{a}{3} + \frac{2}{3}\sqrt{(a^2 - 3b)} \cos\frac{\theta}{3} \tag{9}$$

where:

$$a = K_{d,W} + K_{d,I} + 2P_0 - L_0$$

$$b = K_{d,W}K_{d,I} + K_{d,I}P_0 + K_{d,W}P_0 - (K_{d,W} + K_{d,I})L_0$$

$$c = -K_{d,W}K_{d,I}L_0$$

$$\theta = \arccos\frac{-2a^3 + 9ab - 27c}{\sqrt[2]{(a^2 - 3b)^3}} \qquad (0 < \theta < \pi)$$

*Equations 7 and 8*, with the appropriate substitution of [L] from *Equation 9*, were used as fitting functions for the binding curves for the residues that belong to the WW and PPIase domains, respectively. To keep the fitting functions as simple as possible, we did not combine the $^1$H and $^{15}$N chemical shifts but used them individually to construct the experimental binding curves. The fitting was conducted using IgorPro software (Wavemetrics), with $K_{d,W}$ and $K_{d,I}$ as global parameters and $\Delta_{W,i}$ and $\Delta_{I,i}$ as the local ones.

## Determination of binding affinities using lineshape analysis

$K_d$ values that are determined from the chemical shift analysis become less accurate when the binding regime approaches the 'tight' limit. Therefore, for the $K_d$ values that are smaller than $P_0/5$ (20 μM), we used the lineshape analysis to obtain the information about binding affinity. The residues were selected according to the same criteria as those for the CSP-based analyses. The lineshape analysis was conducted for a total of 241 sites (listed in *Supplementary file 2*) using the software package TITAN (*Waudby et al., 2016*). The [$^1$H, $^{15}$N]-HSQC were processed with exponential line broadening using NMRPipe (*Delaglio et al., 1995*). For each protein-peptide ligand pair, data were fit globally using the 'two-state ligand-binding' model, $P + L \rightleftharpoons PL$, within TITAN. The input parameters were the total protein concentration $P_0$ (100 μM) and the total ligand concentration $L_0$, the latter ranging from 0 to the saturating or near-saturating values. The parameters obtained from the global fits were the dissociation constant $K_d$ and the off-rate constant, $k_{off}$.

## $^1$H-$^1$H exchange spectroscopy

The ability of Pin1 to catalyze the isomerization of the TM was probed using 2D EXSY (*Jeener et al., 1979*). The NMR samples contained 50 μM natural-abundance Pin1 and V5 peptides with concentrations ranging between 1 mM and 2 mM. The EXSY experiments were conducted on the total of six peptides: pV5βII, pV5α, pTMβII, pTMα, pTMα-P640A, and SP-2 (*Supplementary file 1*).

For the -pTPP-containing V5 regions, the maximum mixing time was set to 500 ms. The mixing times $t_{mix}$ for pTMα-P640A were: 20, 35, 50, 75, 100, 140, 200, 280, 400, and 500 ms. The mixing times $t_{mix}$ for the SP-2 were: 10, 25, 30, 40, 70, 100, 150, 200, 300, 400, and 500 ms. The *cis-trans* isomerization of the pThr-Pro bond was monitored by following the intensity of the diagonal and cross-peaks corresponding to the $^1$H$_N$ resonance of Thr641 and Thr638 in the βII and α peptides, respectively. The time dependence of the ratio of intensities of the cis→trans ($I_{ct}$) and diagonal trans-trans peak ($I_{tt}$), $I_{ct}/I_{tt}$, is given by the following equation:

$$\frac{I_ct}{I_tt} = \frac{k_tc[1 - \exp\left(-k_EXt_{mix}\right)]}{k_ct + k_tc\exp(-k_EXt_{mix})}$$
(10)

where $k_{ct}$ and $k_{tc}$ are the forward and reverse rate constants for cis→trans isomerization reaction, $k_{EX}$ is the exchange rate constant that is equal to $k_{ct}+k_{tc}$, and $t_{mix}$ is the mixing time. *Equation (10)* was used to fit the data, with $k_{tc}$ and $k_{ct}$ as the adjustable parameters. The populations of the cis- and trans-conformers of the peptidyl-prolyl pThr-Pro bond were estimated using the SP-2 peptide (LpTPTD), where the cis-cis diagonal peak is well resolved. The populations of the trans- and cis-conformers of the pThr-Pro bond were $I_{tt}/(I_{tt}+I_{cc})$=91% and $I_{cc}/(I_{tt}+I_{cc})$=9%, respectively.

## Determination of the Pin1-pV5βII complex structure using solution NMR spectroscopy

### Sample preparation

We prepared a total of 10 NMR samples for the structural analysis of the Pin1-pV5βII complex. The samples are listed in *Supplementary file 3*, along with the corresponding NMR experiments. The NMR buffer was 10 mM $^2$H$_4$-imidazole at pH 6.6, 100 mM KCl, 1 mM TCEP, 8% D$_2$O, and 0.02% NaN$_3$, except for sample #2* where 100% D$_2$O was used. Samples #1–3 were used to collect the NMR data used for resonance assignments and structure determination. Samples #4–10 provided validation for the assignments. The validation relied on the spectral simplification through the use of single-domain constructs of Pin1 and selectively labeled pTM and pHM regions of pV5βII. All 2D and 3D NMR experiments were acquired at 25°C on the Bruker Avance III HD spectrometers operating at the $^1$H Larmor frequency of 600 MHz or 800 MHz. The temperature was calibrated using deuterated methanol. The chemical shifts were externally referenced to DSS. The NMR data were processed with NMRPipe (*Delaglio et al., 1995*) and analyzed with CcpNmr-Analysis program, version 2.4.2 (*Vranken et al., 2005*).

### NMR structure calculation and refinement

The backbone and sidechain $^1$H resonances of Pin1 complexed to pV5βII were assigned using standard 3D NMR methods (*Supplementary file 3*). Pin1-bound pV5βII $^1$H resonances were assigned using

2D double-filtered NOESY and TOCSY experiments conducted on samples #3 and #5 (*Supplementary file 3*). To facilitate the assignments and overcome the problem of extensive spectral overlap, we prepared NMR samples where the pTMβII and pHMβII peptides were selectively labeled with [U-$^{13}$C,$^{15}$N] amino acids at specific positions (*Supplementary file 1*, peptides #14 and #15; *Supplementary file 3*, samples #6 and #7).

The distance restraints were obtained from the height of the $^1$H-$^1$H cross-peaks in the NOESY spectra. The inter-molecular NOEs were obtained using 3D-$^{15}$N/$^{13}$C-edited $^{13}$C, $^{15}$N-filtered NOESY-HSQC (*Ogura et al., 1996*; *Zwahlen et al., 1997*) on samples #1, #2, #6, and #7; and 2D [F1/F2] $^{13}$C, $^{15}$N-filtered NOESY on sample #3. A total of 2812 (662 of them long-range) intra-Pin1, 241 intra-pV5βII, and 75 inter-molecular Pin1-pV5βII NOEs were used for the structure calculation (*Supplementary file 4*). Hydrogen bonds were identified based on the $^1$H-$^2$D exchange rates of amide $^1$H atoms. Dihedral angles were predicted by the DANGLE routine within the CcpNmr program (*Vranken et al., 2005*) using a complete set of $^{15}$N, $^{13}$C′, $^{13}$C$^α$, $^{13}$C$^β$, $^1$H$^α$, and $^1$H$^N$ chemical shifts.

An ensemble of the Pin1-pV5βII complex structures was calculated using the torsion angle dynamics protocol in CYANA (version 3) (*Güntert and Buchner, 2015*; *Güntert et al., 1997*). Nine hundred random conformers were subjected to 20,000 steps of annealing. The 50 low-energy conformers that had no upper distance violations of >0.2 Å or dihedral angle violations of >5° were used as an input for the refinement procedure implemented in Xplor NIH, version 2.51.5 (*Schwieters et al., 2003*; *Schwieters et al., 2006*). The refinement was conducted in explicit solvent, using the TIP3P water model. The refined ensemble comprising 20 structures was deposited in the PDB under the accession code 8SG2.

## In-cell experiments
### Cell lines
All ex vivo in-cell experiments were performed with STR-authenticated HEK293T cells obtained from the American Type Culture Collection (ATCC catalog number CRL-3216) and their transgenic derivatives. Due to their distinct epithelial morphologies, these cells are not on the list of commonly misidentified cell lines maintained by the International Cell Line Authentication Committee. To maintain cell line integrity, HEK293T cells were not cultured with other cell lines to prevent cross-contamination and cultured cell morphologies were monitored throughout the project. Since both WT and mutant derivative HEK293T cell lines were used in this project, unintended cross-contamination was screened by PCR amplification of distinguishing genomic regions and subsequent DNA sequence analysis of all WT and derivative mutant clonal cell lines to confirm genotype. Cell lines were tested to ensure negative status for mycoplasma contamination using the Universal Mycoplasma Detection Kit (ATCC catalog number 30-1012K).

### Reagents
Chemical reagents were obtained from MilliporeSigma (Burlington, MA, USA) or Thermo Fisher Scientific (Waltham, MA, USA) unless otherwise specified. Disposable plastics, tissue culture dishes, etc. were from Genesee (El Cajon, CA, USA) or VWR (Radnor, PA, USA). Primary antibodies obtained from Cell Signaling Technology (Denvers, MA, USA) included those directed against Pin1 [3722S], beta actin [4970S], PKCα [20565], PKCβ [46809S], and GFP [2555S]. Primary anti-DDK immunoglobulin [TA50011-100] was obtained from OriGene Technologies Inc (Rockville, MD, USA). Secondary goat anti-rabbit IgG (H+L) HRP conjugated antibody was from MilliporeSigma.

### Generation of Pin1 mutant expression plasmids
A Myc/DDK eptiope-tagged Pin1 (NM_006221) Human Tagged ORF Clone (Cat# RC202543) was obtained from OriGene Technologies Inc. The Q5 Site-Directed Mutagenesis Kit (NEB, Ipswich, MA, USA) was used to generate the appropriate Pin1 point mutants: Pin1 W34A, R68A/ R69A, C113S, and W34A/R68A/ R69A. All site-directed mutations were verified by DNA sequence analysis.

The primer sequences for the construction of each point mutant are as follows (sites of mutation are highlighted):

W34A:

 forward primer 5′- CGCCAGCCAGGCCGAGCGGCCCA -3′
 reverse primer 5′- TTAGTGATGTGGTTGAAGTAGTACAC 3′

R68A/R69A:

forward primer 5'- CAGCCAGTCA<u>GCCGCC</u>CCCTCGTCCTGGCG -3'
reverse primer 5'- TGCTTCACCAGCAGGTGC -3'

C113S:

forward primer 5'- GTTCAGCGACA<u>G</u>CAGCTCAGCCA -3'
reverse primer 5'- TGTGAGGCCAGAGACTCAAAG -3'

## Cell culture and plasmid transfections

HEK293T cells were cultured in a humified incubator at 37°C and 5% $CO_2$ in high-glucose Dulbecco's Modified Eagle Medium plus sodium pyruvate (Genesee) and supplemented with 10% FBS and penicillin (100 units/ml)/streptomycin (100 µg/ml). After trypsinization to release adherent cells, viable cell counts and cell size were determined by trypan blue staining (Invitrogen, Thermo Fisher Scientific) followed by passage through a Countess II automated cell counter (Thermo Fisher Scientific) according to the manufacturer's instructions. Plasmid transfections were performed using Lipofectamine LTX with Plus reagent (Thermo Fisher Scientific) according to the manufacturer's instructions. Stable expression lines were generated by transfection followed by serial selection with 2 mg/ml geneticin (Gibco).

## Cell lysate preparation and Pin1 immunoblotting

Expression of Pin1, its corresponding variants, and PKCα levels were monitored by immunoblotting. Cells were solubilized in lysis buffer (BB150: 50 mM Tris pH 7.6, 150 mM NaCl, 0.2% CHAPS, 10 mM EDTA) on ice and centrifuged at 21,000×$g$ for 5 min at 4°C to remove insoluble materials. Protein in the soluble fractions was quantified using the colorimetric bicinchoninic acid assay (Pierce) and samples were prepared with 80 µg of total lysate protein in 1× Laemlli buffer (62.5 mM Tris-HCl, 2% SDS, 25% glycerol, and 0.01% bromophenol blue, pH 6.8). Samples were subsequently boiled for 5 min and resolved by SDS-PAGE using 15% acrylamide gels (120 V constant voltage for 80 min) and transferred onto 0.2 µm nitrocellulose membranes (Bio-Rad) at 350 mA (constant current) for 1 hr. Following transfer, membranes were blocked for 1 hr in 5% milk powder (wt/vol) in 20 mM Tris pH 7.6, 150 mM NaCl, 0.1% Tween 20 (TBST), and incubated at 4°C overnight with primary antibodies at a 1:1000 dilution in Tris-buffered saline (pH 7.6), 0.1% Tween 20, 5% BSA (wt/vol) with rocking. Following incubation, blots were washed three times for 10 min each in TBST and incubated with goat anti-rabbit secondary antibody coupled to horseradish peroxidase (MillliporeSigma; 1:10,000 dilution) in TBST/5% BSA for 1 hr at room temperature. Following three additional 10 min washes with TBST, blots were developed using SuperSignal West Femto Maximum Sensitivity Substrate (Thermo Fisher Scientific) and analyzed by densitometry using a Molecular Imager ChemiDoc XRS+ system (Bio-Rad Laboratories, Hercules, CA, USA). Protein levels were normalized to total protein load and verified using actin as loading control.

## Quantification of PKCα levels

PKCα levels were assessed in cells seeded at 1.5×10⁶ cells per 35 mm dish 24 hr prior to analysis. Cells from three dishes were pooled per sample. Cells were seeded in complete media containing 10% FBS unless otherwise stated. Following incubation, cells were washed with phosphate-buffered saline and snap-frozen in liquid nitrogen prior to analysis.

## siRNA-mediated knockdown of Pin1

siRNA transfections were performed using Dharmafect (Dharmacon/Horizon) according to the manufacturer's instructions in antibiotic free growth media. HEK293T cells were incubated with siRNA complexes overnight and complete media were exchanged the following day. Optimal Pin1 knockdown was determined to be at 72 hr post-transfection. All transfection complexes for both sets of reagents were prepared in optimum media lacking FBS or antibiotics. The ON-TARGET plus Human Pin1 (5300) siRNA-SMART pool (Dharmacon/Horizon) sequences were as follows: J-003291-10 5'-GCUCAGGCCGAGUGUACUA-3', J-003291-11 5'-GAAGACGCCU CGUUUGCGC-3', J-003291-12 5'-GAAGAUCACCCGGACCAAG-3', and J-003291-13 5'-CCAC AUCACUAACGCCAGC-3'.

## Isolation of a clonally derived Pin1 null cell line

To generate clonal cell lines lacking Pin1 protein, HEK293T cells were sequentially transfected with two CRISPR/Cas9 plasmids driving expression of guide RNAs that specifically target exon2 of the *Pin1* structural gene (MilliporeSigma product numbers and targeting sequences: CRISPRD HSPD0000031439 5'-GA GAAGATCACCCGGACCA-3', CRISPRD HSPD0000031440 5'-TAACGCCA GCCAGTGGG AG-3'). Transfected cells were flow-sorted on the basis of GFP fluorescence on a three-laser (405 nm, 488 nm, 640 nm) Beckman Coulter Moflo Astrios high-speed cell sorter at a rate of <1000 events/s. Cell debris were gated and eliminated from the bulk sort. GFP was detected using a 664/22 nm bandpass filter, and Cas9-GFP positive cells were bulk-sorted into 1.5 ml tube using the 'Purify mode' routine with a drop envelope of 1–2 droplets.

Following an initial round of sorting, cells were allowed to recover, and the bulk-sorted population was re-transfected with the second CRISPR/Cas9 plasmid and re-sorted as described. Those second-round bulk-sorted cells were tested to assess overall Pin1 expression within the heterogeneous cell population by immunoblotting for Pin1 antigen. In procedures where the bulk population expressed ≤50% of WT Pin1 levels using total protein to normalize data, individual cell lines were developed from the bulk-sorted population by limiting serial dilution and expansion of single cells into clonal cell populations. Clonally derived cell lines lacking detectable Pin1 antigen were identified by immuno-blotting with anti-Pin1 immunoglobulin.

Clonally derived cell lines were verified by sequencing of individual *Pin1* alleles. Total genomic DNA was isolated from each cell line using the GenElute Mammalian Genomic Miniprep Kits (MilliporeSigma). The *Pin1* coding region from exons 2 and 3 was recovered by amplifying each of these coding exons individually via polymerase chain reaction using the Phusion-Plus high-fidelity proof-reading enzyme (ThermoFisher Scientific). Primers used for amplification of each exon were as follows:

> Exon 2 -- Forward 5'-CTGGGAGCACAACCCTAGC-3'
> Reverse 5'-AGGTCATGCACTGGCGTTTT-3
> Forward 5'-GGGAGCACAACCCTAGCTG-3
> Reverse 5'-CTACAAAGGCTCACCTGGGA-3
> Exon 3 -- Forward 5'-CTGGCACTCCCATTCCGTTC-3
> Reverse 5'-CCTGCCATGTCATCTGTCCC-3
> Forward 5'-ACTCCCATTCCGTTCCATGTC-3
> Reverse 5'-CCCTGCCATGTCATCTGTCC-3
> Exon 4 -- Forward 5'-CAGGTCAGATGCAGAAGCCAT-3
> Reverse 5'-CCACGACATCTTCCCCACTAT-3
> Forward 5'-AGGTCAGATGCAGAAGCCATT-3
> Reverse 5'-GATCCCCTCCCCACGACATC-3

PCR products were extended by *Taq* polymerase-driven poly-deoxyadenosine tailing in 30 min. Tailed PCR products were analyzed by agarose gel electrophoresis alongside products generated from a WT HEK293T cell control. Bands were excised and gel purified using a Qiaquick Gel Extraction Kit (QIAGEN). Purified DNA fragments were ligated into the pGEMt-easy plasmid (Promega) and transformed into DH5α bacteria selected on standard LB agar plates containing ampicillin (100 µg/ml), X-gal (50 µg/ml), and IPTG (1 mM). Blue/white colony screening was employed to identify those transformants harboring plasmids containing inserts (white colonies). Plasmids were prepared from individual colonies using the QIAprep Spin Miniprep Kit and individual allele sequences were obtained by DNA sequencing using the T7 primer to program the sequencing reaction.

## Production of stable transgenic cell lines expressing Pin1 variants

To generate cell lines expressing WT or mutant Pin1 proteins as sole sources for Pin1 activity, HEK293T Pin1 KO cells (clone C3) were transfected with WT Pin1, the W34A, R68A/R69A, W34A/R68A/R69A, or C113S transgenes tagged with the myc/DDK epitopes. The epitope tags were engineered at the Pin1 C-terminus using the following DNA sequence (myc and DDK coding sequences underlined):

5'- ACG CGT ACG CGG CCG CTC <u>GAG CAG AAA CTC ATC TCA GAA GAG GAT CTG</u> GCA GCA AAT GAT ATC CTG GAT TAC AAG <u>GAT GAC GAC GAT AAG</u> GTT TAA -3'

Twenty-four hours post-transfection, an aliquot of the transfected cells was harvested to generate lysates from which successful transfection and protein expression were assessed by

immunoblotting for Pin1 antigen. The remaining cells were cultured in high-selection media (2 mg/ml geneticin) and grown at low density, with repeated passage and frequent media changes, over a period of 3 weeks. Pin1 expression was subsequently re-examined by immunoblotting for Pin1 antigen and verified stable lines were consistently maintained under low-selection conditions (1 mg/ml geneticin).

## Cell area and cell viability measurements by flow cytometry

Cells were loaded into the Image Stream X Mark II imaging flow cytometer (Cytek/Amnis Freemont, CA, USA) to measure cell area and viability. Cells were analyzed at a concentration of $1 \times 10^6$ cells in 50 µl of FACS buffer (phosphate-buffered saline, 0.5% bovine serum albumin) supplemented with eBioscience propridium iodide (Invitrogen) to a final concentration of 2 µg/ml. All data from the Image Stream were acquired using the Inspire program (version 201.1.0.765; Cytek/Amnis Freemont, CA, USA). The acquisition parameters used were 40× objective, medium speed, 488 nm wavelength, laser power 100 mW, SSC laser power 2.81 mW. The scatter channel was set for channel 6, the bright-field channels were set to channels 1 and 9. The core stream size was set to 6 µm, and 10,000 events of focused, live, single cells were collected. The focus was determined by plotting the Gradient RMS values for channel 1 bright field in a histogram and selecting cells that possessed a gradient RMS value of ≥40. Single cells were identified in the focused cells by plotting the aspect ratio of the channel 1 bright field (y-axis) versus channel 1 bright-field area (x-axis). Single cells reside in the range of 0.6–1 in the aspect ratio and between 0 and $1 \times 10^3$ µm² range in the area of channel 1. Cell viability was determined by plotting the intensity of channel 5 (propidium iodide) on the y-axis versus intensity of channel 6 (side scatter). The data were analyzed using the acquisition gates and settings in the IDEAS 6.3 (Cytek/Amnis Freemont, CA, USA) data analysis software package. The median of the cell area and % viability parameters were calculated for each sample.

## Acknowledgements

This work was supported by grants NIH RO1 GM108998 and Robert A Welch Foundation A-1784 to TII. YY was an American Heart Association predoctoral fellow (award no. 14PRE20380475). MIM and VAB were supported by grants NIH R35 GM131804 and award BE-0017 from the Robert A Welch Foundation to VAB. We acknowledge Gus Wright (Texas A and M Veterinary School Flow Cytometry Core) for assistance in the cell sorting component of the PIN1 gene editing experiments and with cell viability and cell area measurements. We acknowledge the Protein Chemistry Laboratory (Department of Biochemistry and Biophysics, Texas A&M University) for assistance with mass spectrometry experiments.

## Additional information

### Funding

| Funder | Grant reference number | Author |
|---|---|---|
| National Institute of General Medical Sciences | RO1GM108998 | Tatyana I Igumenova |
| Welch Foundation | A-1784 | Tatyana I Igumenova |
| National Institute of General Medical Sciences | R35GM131804 | Vytas A Bankaitis |
| American Heart Association | 14PRE20380475 | Yuan Yang |
| Welch Foundation | BE-0017 | Vytas A Bankaitis |

The funders had no role in study design, data collection and interpretation, or the decision to submit the work for publication.

## Author contributions
Xiao-Ru Chen, Data curation, Formal analysis, Validation, Investigation, Visualization, Writing – original draft, Writing – review and editing; Karuna Dixit, Data curation, Formal analysis, Validation, Investigation, Visualization, Writing – original draft; Yuan Yang, Conceptualization, Formal analysis, Investigation, Visualization, Writing – review and editing; Mark I McDermott, Formal analysis, Investigation, Visualization, Writing – review and editing; Hasan Tanvir Imam, Investigation, Visualization; Vytas A Bankaitis, Formal analysis, Supervision, Funding acquisition, Methodology, Writing – original draft, Writing – review and editing; Tatyana I Igumenova, Conceptualization, Formal analysis, Supervision, Funding acquisition, Visualization, Methodology, Writing – original draft, Project administration, Writing – review and editing

## Author ORCIDs
Xiao-Ru Chen (ID) https://orcid.org/0000-0002-2051-889X
Karuna Dixit (ID) http://orcid.org/0000-0003-3583-2512
Yuan Yang (ID) http://orcid.org/0000-0002-9538-6369
Vytas A Bankaitis (ID) http://orcid.org/0000-0002-1654-6759
Tatyana I Igumenova (ID) https://orcid.org/0000-0003-3772-7484

## Decision letter and Author response
Decision letter https://doi.org/10.7554/eLife.92884.sa1
Author response https://doi.org/10.7554/eLife.92884.sa2

---

# Additional files

## Supplementary files
• Supplementary file 1. Properties of the 18 PKC C-terminal-derived peptides used in this study. All peptides have acetylated N-termini and amidated C-termini. TM and HM stand for the turn and hydrophobic motifs, respectively. The phosphorylated Thr of the TM and phosphorylated Ser of the HM are shown in red. Peptides having 'V5' in their name contain both TM and HM. Peptides starting with 'p' indicate that the peptide is phosphorylated at either one or both motifs, the latter only for the 'V5' peptides.

• Supplementary file 2. List of binding experiments carried out in this study, with the corresponding values of the dissociation constants $K_d$ obtained from the chemical shift binding curves and/or lineshape analysis.

• Supplementary file 3. List of the NMR samples and experiments for the structure determination of the complex. Sample 2* was prepared in the buffer containing 100% $D_2O$.

• Supplementary file 4. NMR restraints statistics for the CYANA structure calculation.

• MDAR checklist

## Data availability
The atomic coordinates, restraints, chemical shifts, and peak lists for the Pin1::pV5bII complex are deposited in Protein Data Bank (accession code 8SG2) and Biological Magnetic Resonance Data Bank (accession code BMRB-31080).

The following datasets were generated:

| Author(s) | Year | Dataset title | Dataset URL | Database and Identifier |
|---|---|---|---|---|
| Dixit K, Chen XR, Igumenova TI | 2023 | A novel bivalent interaction mode underlies a non-catalytic mechanism for Pin1-mediated Protein Kinase C regulation | https://www.rcsb.org/structure/unreleased/8SG2 | RCSB Protein Data Bank, 8SG2 |
| Dixit K, Chen XR, Igumenova TI | 2023 | Data from: A novel bivalent interaction mode underlies a non-catalytic mechanism for Pin1-mediated Protein Kinase C regulation | https://bmrb.io/data_library/summary/index.php?bmrbId=31080 | Biological Magnetic Resonance Data Bank, 31080 |

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
