## [Editor Report]

Pin1 is an essential prolyl cis/trans isomerase which has attracted considerable attention as this enzyme family is implicated in cancer and neurodegenerative diseases. Here, authors provide compelling evidence that Pin1 modulates the activity of an important cell signaling kinase, Protein Kinase C by a non-catalytic mechanism. This result suggests a new mode of Pin1 action, regulating the cellular stability of the kinase via a chaperone-like activity.

---

## [Decision Letter]

**Decision letter after peer review:**

Thank you for submitting your article "A novel bivalent interaction mode underlies a non-catalytic mechanism for Pin1-mediated Protein Kinase C regulation" for consideration by *eLife*. We apologize the unusual delay and thank you for the patience. Your article has been reviewed by 2 peer reviewers, and the evaluation has been overseen by a Reviewing Editor and Jonathan Cooper as the Senior Editor.

Essential revisions (for the authors):

Overall, reviewers are enthusiastic about your work and provided helpful comments that can further strengthen the manuscript. They also suggested a few additional experiments. I encourage you to consider those suggestions or to justify your approaches and results. Please, provide point-by-point responses to all reviewers' comments.

1) Address the possibility that the orientation of the PPIase/WW domain interface and the conformation of the bound V5-derived peptides are dynamic rather than static.

2) Provide further explanations on the cell-based assay results, regarding the expression levels of PKC in the original WT and KO cell lines.

3) Emphasize the novelty of the Pin1-V5 peptide complex structure. Yeh et al. (JBC 2006) reports a potential noncanonical binding mode in the PIN1-cyclinE interaction.

4) Some CSP pattern may reflect allosteric or dynamic effects and only indirectly reflect binding. Also, further evidence of unidirectionality is required. NOE experiments for structure determination may help. Employing alternative methods for measuring Kd values (ITC, MST, etc.) would be worthwhile to consider.

5) A more in-depth analysis of the binding data based on the kcat/Km values will be useful to provide further insights into the Pin1-client interaction and the contribution of phosphate groups to the binding. In addition, EXSY NMR may miss some cis-trans isomerization events that can possibly occur beyond the time scale of the experiment. Please, address these points.

6) Provide mass spectrometry data of the phosphorylated peptides in the SI.

*Reviewer #1 (Recommendations for the authors):*

As the structure (8SG2.pdb) is still on hold, it is harder to evaluate it in detail. I recommend the authors release it.

The connection between the structural model and the in vivo data should be strengthened.

*Reviewer #2 (Recommendations for the authors):*

The following recommendations would greatly improve the quality and impact of the manuscript and certainly make it suitable for endorsement by *eLife*:

– Bivalent interaction: My biggest criticism is that all the data supporting the unidirectional bivalent interaction model are based only on cherry-picked residue specific Kds from NMR titrations. This can be misleading because chemical shifts may report also on allosteric or dynamic effects. The conclusions should be orthogonally supported by global Kd determination from e.g. ITC, MST, etc. The CSP patterns are not sufficient to prove unidirectionality, the mono-phosphorylated peptide could swap and bind to both sites, a FRET reporter assay or PRE experiments would decipher this.

– Enzymatic activity: The absence of enzymatic activity of Pin1 towards PKC is central to the paper and should be very carefully addressed. To me, the best hint for this is that expression of a catalytically dead mutant rescues PKC levels in Pin1 knock-out cells. However, the in vitro data are not as clear, actual rates should be quantified and an estimate of the kcat/Km should be given even if it is very low. It would be interesting to decipher the contribution of kcat and Km to the lower activity of Pin1 towards S/TPP motifs. All enzyme assays are based on EXSY NMR which is only sensitive to a narrow timescale. It would be advisable to probe other timescales using real-time NMR, and to quantify cis/trans populations based on Pro Cb-Cd 13C shifts.

---

## [Author Response]

Essential revisions (for the authors):Reviewer #2 (Recommendations for the authors):The following recommendations would greatly improve the quality and impact of the manuscript and certainly make it suitable for endorsement by eLife:– Bivalent interaction: My biggest criticism is that all the data supporting the unidirectional bivalent interaction model are based only on cherry-picked residue specific Kds from NMR titrations. This can be misleading because chemical shifts may report also on allosteric or dynamic effects. The conclusions should be orthogonally supported by global Kd determination from e.g. ITC, MST, etc. The CSP patterns are not sufficient to prove unidirectionality, the mono-phosphorylated peptide could swap and bind to both sites, a FRET reporter assay or PRE experiments would decipher this.

We respectfully disagree with the reviewer’s assessment that “all the data supporting the unidirectional bivalent interaction model are based only on cherry-picked residue specific Kds from NMR titrations”.

The NMR structure of Pin1::pV5bII complex (Figure 6 and Figure 5—figure supplement 2) is the ultimate proof of the bivalent unidirectional binding mode. The structure was calculated based on the experimental restraints, including 75 inter-molecular NOEs across the Pin1-pV5bII binding interface (see Supplementary File 1). To illustrate this point, we have included several inter-molecular ^1^H-^1^H strips as panels A and B of Figure 5—figure supplement 2. The CSP/Kd data of Figure 4 are entirely consistent with and fully support the unidirectional bivalent binding mode.

With respect to the monophosphorylated peptides, we provide evidence for unambiguous determination of the binding modes by adding new figure panels, Figure 5A-D. The chemical shift data and the correlation plots unambiguously show the unidirectional bivalent binding mode for the mono-phosphorylated V5-pTM-HM segment (Figure 5A-B), and a lack of bivalent unidirectional interactions for the V5-TM-pHM monophosphorylated region (Figure 5C-D). We have also added cartoon representations of the binding modes in Figure 5E-H. We hope these revisions enhance the clarity of the representation and eliminate confusion regarding the binding modes.

The reviewer suggests to conduct FRET and PRE experiments to provide additional evidence for unidirectionality (we assume this comment applies to the mono-phosphorylated V5-pTM-HM region, the data for which are now presented in Figures 5A,B,F). Indeed, we have considered these types of experiments at the initial stages of the project. However, we ruled them out because of their perturbing nature -- as applied to our system. FRET/PRE would require an introduction of at least three mutations into Pin1: mutation of two native Cys residues (one of them, Cys113, is a catalytic residue that lines the PPIase binding site) and introduction of another solvent-exposed Cys for the modification with a bulky nitroxide label or a fluorophore. Modification of the C-term PKC region raised similar concerns related to mutagenesis and introduction of bulky probes. In our assessment, the structurally non-perturbing approach implemented in the current study is superior and produces clearly interpretable data as shown in Figure 5.

Regarding “cherry-picking” of residues and K_d_ determination:

In selecting Pin1 residues for the binding analyses we followed rigorous procedures accepted in the NMR field, and absolutely no “cherry-picking” was done. Moreover, the reported Kd values are not residue-specific but are the results of global fits (see Supplementary File 2). We also emphasize that we did not rely just on the CSP-based binding curves but also conducted lineshape analysis that includes fitting parameters such as populations of bound and unbound species. To clarify these points and emphasize that we observed no indirect effects (allosteric/dynamic), we have expanded the Methods section to provide more details on the procedures of the binding analyses on page 33 [CSP-based binding curves] and pages 35-36 [lineshape analysis].

Due to space limitations, the main figures show binding data for selected residues with global fits. All results of the binding analyses (459 experimental binding curves with fits and 241 peaks with lineshape fits, see Supplementary File 2) are available from the authors upon request. Full data or their subset can be included in the Supporting Information if the reviewer and editors deem it essential.

The reviewer suggests using orthogonal approaches such as ITC or MST. Neither ITC nor MST can provide domain-specific information for full-length Pin1, which was absolutely essential for us to be able to dissect the bivalent mode. We do not see what additional information ITC or MST can provide for our study -- unless the reviewer can point specifically to where they think our NMR-based approach is inadequate or deficient.

– Enzymatic activity: The absence of enzymatic activity of Pin1 towards PKC is central to the paper and should be very carefully addressed. To me, the best hint for this is that expression of a catalytically dead mutant rescues PKC levels in Pin1 knock-out cells. However, the in vitro data are not as clear, actual rates should be quantified and an estimate of the kcat/Km should be given even if it is very low. It would be interesting to decipher the contribution of kcat and Km to the lower activity of Pin1 towards S/TPP motifs. All enzyme assays are based on EXSY NMR which is only sensitive to a narrow timescale.

Regarding kcat/Km measurements: The chymotrypsin-coupled chromophore (p-nitroaniline) assay is currently the only available method to determine the kcat/Km for Pin1-catalyzed reactions. A significant limitation of the assay (pointed out by other researchers in the field, see Greenwood et al., 2011, Complete determination of the Pin1 catalytic domain thermodynamic cycle by NMR lineshape analysis. Journal of biomolecular NMR 51(1-2), 21) is that chymotrypsin only cleaves at the C-terminal region of an aromatic residue that follows a residue that is preceded by a trans peptide bond. This assay is incompatible with our system that has a double-Proline motif, leaving the ^1^H-^1^H EXSY experiment the only available route to assay Pin1 catalytic activity.

Regarding EXSY experiments: EXSY NMR is sensitive to the timescale range that covers ca. 3 orders of magnitude, from 0.1 s^-1^ to 100 s^-1^. We have included this statement with appropriate citations on page 7. We have also included an estimate of the exchange rate (< 0.1 s^-1^) in our experiments on pages 7-8.

It would be advisable to probe other timescales using real-time NMR, and to quantify cis/trans populations based on Pro Cb-Cd 13C shifts.

Given that the equilibrium between cis- and trans- isomers is established in solution prior to the addition of Pin1, and that addition of Pin1 does not change the populations, it is not feasible to monitor this reaction using real-time NMR.

Quantifying cis-trans populations using the intensities of Cb/Cg cross-peaks requires incorporation of isotopic labels into pV5bII, as the ^1^H-only based 2D experiments are not adequate for this purpose. Recombinant expression of isotopically enriched pV5bII is not feasible because of phosphorylation, while chemical synthesis of pV5bII with labeled amino acids is prohibitively expensive. Quantification of populations, while providing generally useful information, has no bearing on the conclusions of this work.